# Heterogeneous Ru Catalysts as the Emerging Potential Superior Catalysts in the Selective Hydrogenation of Bio-Derived Levulinic Acid to γ-Valerolactone: Effect of Particle Size, Solvent, and Support on Activity, Stability, and Selectivity

**Mulisa Maumela** [1,*], **Sanette Marx** [2] **and Reinout Meijboom** [3,*]

1    Department of Mathematics, Science and Technology Education, University of Limpopo, Private Bag x1106, Sovenga 0727, South Africa

2    DST/NRF Research Chair in Biofuels and Other Clean Alternative Fuels, Centre of Excellence in Carbon-Based Fuels Faculty of Engineering, North-West University, Potchefstroom 2531, South Africa; Sanette.Marx@nwu.ac.za

3    Department of Chemical Sciences, University of Johannesburg, P.O. Box 524, Auckland Park, Johannesburg 2006, South Africa

*    Correspondence: mulisa.maumela@ul.ac.za (M.M.); rmeijboom@uj.ac.za (R.M.); Tel.: +27-15-268-3861 (M.M.); +27-11-559-2367 (R.M.)

**Abstract:** Catalytic hydrogenation of a biomass-derived molecule, levulinic acid (LA), to γ-valerolactone (GVL) has been getting much attention from researchers across the globe recently. This is because GVL has been identified as one of the potential molecules for replacing fossil fuels. For instance, GVL can be catalytically converted into liquid alkenes in the molecular weight range close to that found in transportation fuels via a process that does not require an external hydrogen source. Noble and non-noble metals have been used as catalysts for the selective hydrogenation of LA to GVL. Of these, Ru has been reported to be the most active metal for this reaction. The type of metal supports and solvents has been proved to affect the activity, selectivity, and yields of GVL. Water has been identified as a potential, effective "green" solvent for the hydrogenation of LA to GVL. The use of different sources of $H_2$ other than molecular hydrogen (such as formic acid) has also been explored. In a few instances, the product, GVL, is hydrogenated further to other useful products such as 1,4-pentanediol (PD) and methyl tetrahydrofuran (MTHF). This review selectively focuses on the potential of immobilized Ru catalysts as a potential superior catalyst for selective hydrogenation of LA to GVL.

**Keywords:** ruthenium; heterogeneous; levulinic acid; hydrogenation; γ-valerolactone

## 1. Introduction

One of the biggest challenges facing the world recently is the production of sufficient energy as per world population demand. For example, one-third of the energy produced throughout the world is utilized in the transportation industry. Most of this energy is solely derived from fossil resources, i.e., crude oil. As a result, that these fossil resources can deplete some time, there is an urgent need to find alternative, renewable sources of energy. Moreover, it is important to develop industrial production processes that are sustainable and environmentally friendly (greener). Presently, the first generation of biofuels can be produced from sugars, starches, and vegetable oil. However, biofuels produced from these biomass sources face direct competition from the food industry. As a consequence, it is unlikely that enough volume of biofuels as per global demand can be produced from them. Lignocellulosic biomass, which can be derived from sugars, grass, wood, and other agricultural waste, has been identified as a potential renewable replacement for fossil fuels [1–3]. The use of these biomass-derived molecules as an

alternative industrial feedstock to produce biofuels, and other valuable chemicals has been found to have combined advantages in reducing environmental challenges like global warming (reduction of $CO_2$ emission) and economic benefits since the used biomass is renewewable.

Since a more significant portion of this cellulose is a constituent of plant cell walls, it faces no competition from the food industry. Hemicellulose, and lignin are the three main components constituting lignocellulosic biomass. Of these, cellulose, which accounts for 30–50 wt% of lignocellulosic biomass, can be converted into glucose via chemical or enzymatic hydrolysis [4,5]. It is from this formed glucose that some important platform chemicals such as 5-hydroxymethylfurfural (HMF), levulinic acid (LA), and formic acid (FA), ethanol, and liquid fuels can be produced (see Figure 1) [4,6,7]. The other two components, hemicellulose, and lignin each, independently, account for 15–30 wt% of lignocellulosic biomass [8]. Hemicellulose is an amorphous polymer comprised of $C_5$ and $C_6$ sugars while lignin on the other side, is also an amorphous polymer that is rich in aromatic monomers. Additionally, lignin also has been reported to be a potential source for biofuels and other chemicals of great value [9]. However, the complexity of its structure and non-uniformity of its composition render it less attractive as compared to the other two components in this regard. Additionally, about 170 tons of these biomass materials are produced naturally via photosynthesis.

The catalytic conversion of sugars to HMF, LA, and FA has been getting a lot of attention from many scientists in the last few decades [10,11]. For instance, Qi et al. reported the catalytic conversion of sugars (fructose, glucose, and sucrose) into LA, FA, and HMF via hydrolysis [12,13]. The conversion of various sugars was carried out by adding a desired amount of sugars into the mixture containing a diluted acid (dilute $H_2SO_4$) and a solvent. The mixture was then heated at elevated temperatures (130 °C) until the decomposition of the sugar is complete. Decomposition products were analyzed by NMR. Various solvents, such as DMSO, tetrahydrofuran (THF), MeCN, acetone, γ-valerolactone (GVL), or dichlomethane (DCM) were also investigated in the decomposition of fructose. It was found that DMSO and THF showed higher selectivity for the formation of HMF. However, the use of THF has environmental issues due to the formation of peroxides when exposed to air. Using GVL as a solvent showed selectivity towards HMF of about 70%.

The concentration of acid catalyst ($H_2SO_4$) in the GVL solvent was found to affect the selectivity of the products. For instance, no HMF or LA could be detected when the acid concentration is below $10^{-4}$ mol/L. The maximum conversion for HMF and LA (58% and 13%, respectively) was observed when the acid concentration of 0.1 mol/L was used. However, as the concentration of the acid was increased to 0.6 mol/L, a decrease (by 2%) in the formation of HMF was observed with that of LA raising to 62%. This was, therefore, the indication that the product distribution in GVL is sensitive to the acid concentration. Moreover, GVL proved to be a tunable solvent for the conversion of carbohydrates to either HMF, LA, and FA. The HMF can also be converted further into LA or FA in the presence of acid $H_2SO_4$ and the GVL solvent at 130 °C. Likewise, the LA yield was observed to rapidly increase as the concentration of the acid is increased. The conversion of carbohydrates into LA or FA has been reported elsewhere [13]. The discussion of this review will however put the focus on the supported Ru-catalyzed conversion of LA to GVL via hydrogenation reaction.

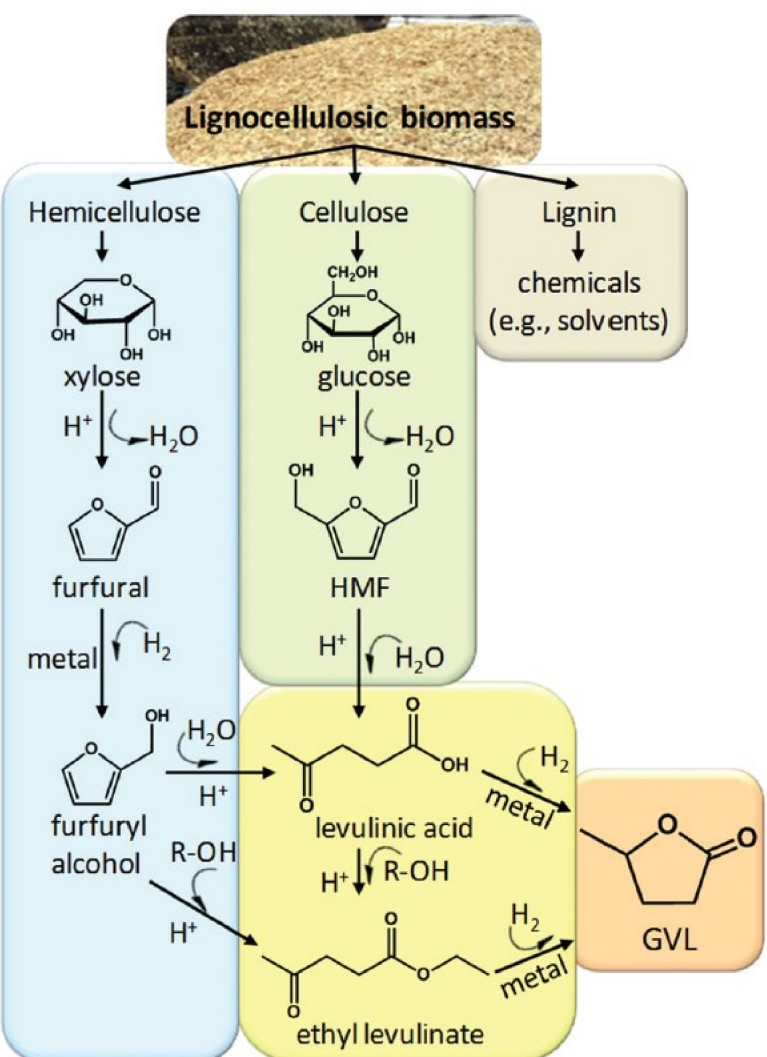

**Figure 1.** Fractionation of lignocellulosic biomass and the reaction pathway to produce γ-valerolactone (GVL) from hemicellulose and cellulose. Reproduced from ref [14] with permission from The Royal Society of Chemistry.

One of these essential platform chemicals, LA, has been named among the top 12 essential platform chemicals US Department of energy [15,16]. LA can undergo catalytic hydrogenation to produce γ-valerolactone (GVL) [17]. However, the development of suitable economical production technology for large-scale production of GVL direct from carbohydrates remains an ongoing challenge. The four consecutive steps for the reaction process (conversion of carbohydrates to GVL) involve the acid-catalyzed dehydration of carbohydrates to HMF [13,18–22], the acid-catalyzed hydration of HMF to LA and FA [12,23], and catalytic hydrogenation of LA to 4-hydroxyvaleric acid (4-HVA) and a subsequent ring closure via dehydration to form GVL [1]. It is noteworthy to mention that other valuable chemicals can be obtained by direct catalytic conversion of LA [1,24,25].

For example, Luo et al. reported selective, one-pot conversion of LA to pentanoic acid (PA) using bifunctional zeolite supported Ru catalysts (Ru/H-ZSM5) [25]. In their study, a series of Ru/H-ZSM5 catalysts were prepared by varying metal precursors, Si/Al ratio via wet impregnation method. The average sizes of the particles of the Ru/H-ZSM5 catalysts were determined by TEM to be in the range of 1.7–4.9 nm. The Ru/H-ZSM5 catalyst was prepared using the $Ru(NH_3)_6Cl_3$ and $NH_4^+$-ZSM5 (Ru/H-ZSM5(NH_3, A, 11.5)) was found to have the smallest particle size (1.7 nm) compared to other Ru/H-ZSM5 catalysts. A higher PA yield (91.3%) was achieved when this catalyst was used under standard

conditions (200 °C, $H_2$ = 40 bar, 10 h). The improved activity of the Ru/H-ZSM5($NH_3$, A, 11.5) catalyst in the direct conversion of LA to PA was attributed to improved dispersion of small Ru particles and the presence of accessible acidic sites required for this type of reaction. In another related study, LA was converted to 5-nonanone over Pd/$Nb_2O_5$ catalysts via the intermediate formation of GVL [24]. Production of liquid alkenes that can be utilized for transportation fuels from LA has been reported by Bond et al. [2]. This was achieved by steps involving ring-opening of GVL to form an isomeric mixture of PA in the presence of HZSM-5 and water as an internal source of hydrogen. PA subsequently undergoes decarboxylation to produce butane. In the presence of the catalyst, butane can undergo oligomerization to form a mixture of oligomers Another interesting study on the conversion of LA to valeric acid (VA) via selective electrocatalytic hydrogenation using different metallic electrodes (Pb, Zn, Co, Pt, and Cu) has reported by Du et al. [26]. Of these metallic electrodes, Pb was found to have higher LA conversions and improved selectivity towards GVL. Possible schemes for the catalytic conversion of LA to values added chemicals, biofuels, and solvents are shown in Figure 2 The discussion in this review, however, will be limited to the supported Ru-based catalytic conversion of LA to GVL.

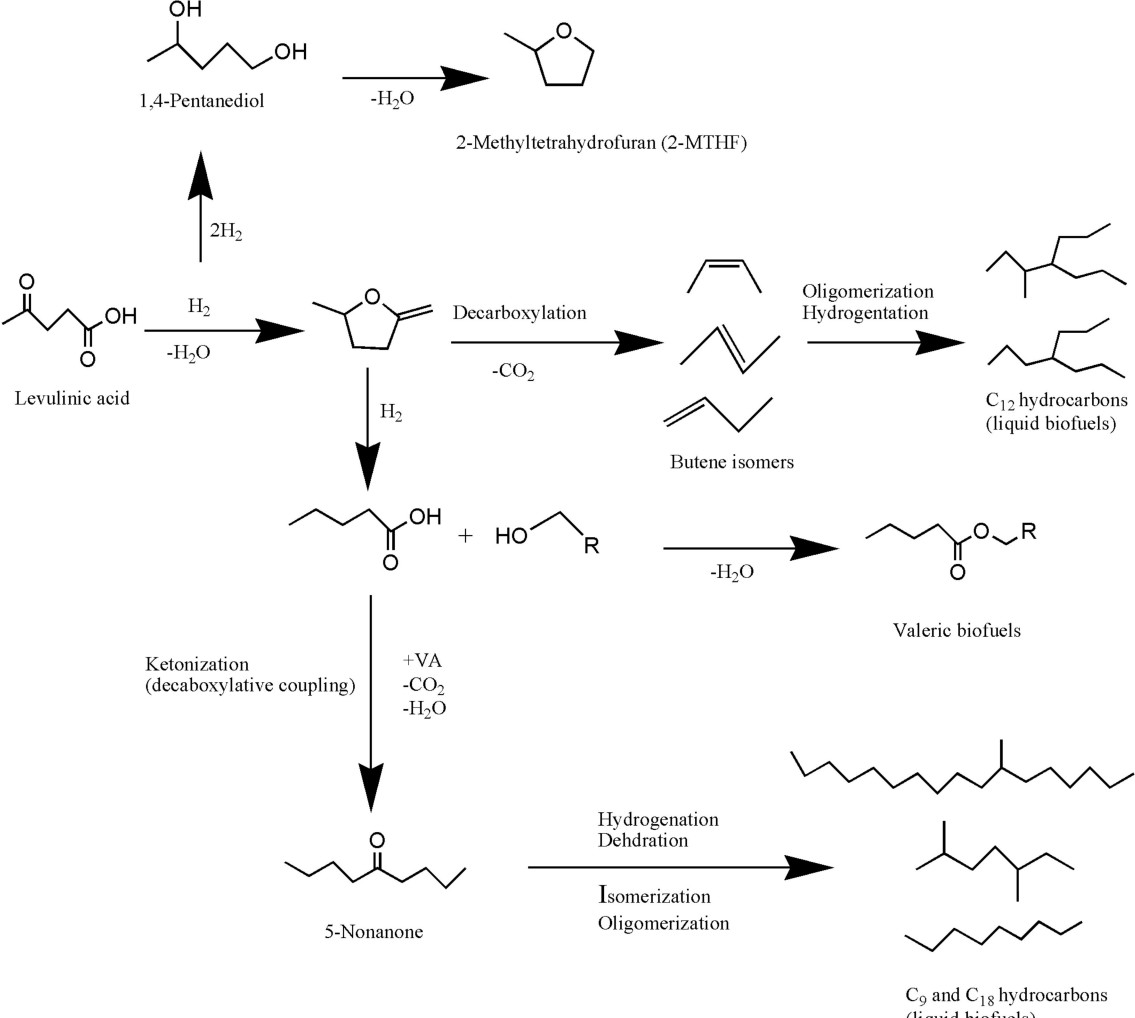

**Figure 2.** Different routes for the production of chemicals, biofuels, and solvents via catalytic hydrogenation of LA.

The versatile molecule GVL has been identified as a potential sustainable liquid for renewable energy production [27,28]. Interestingly, GVL can further be converted via catalytic hydrogenation to produce other important chemicals such as 1,4-pentanediol (PD), methyl tetrahydrofuran (MTHF), and pentanoic acid [3,13,29] (see Figure 3). Alternatively,

GVL can be used directly as a fuel additive with similar or even better properties than those of ethanol or liquid alkenes in biofuels [2,28,30,31]. This is because water can easily be separated from GVL as compared to ethanol. Most importantly, GVL has been identified as a petrol additive [27] that can potentially greatly reduce the emission of carbon monoxide [32], a sustainability issue faced by large cities around the globe. Additionally, it is also used as a green solvent, food additives, or perfume [33,34]. For instance, Al-Shaal et al. reported the solvent-free hydrogenation of GVL to 2-MTHF in the presence of a Ru/C catalyst, achieving an MTHF yield of 43%. The conversion of GVL into liquid alkenes with an appropriate molecular weight range to be used for transportation fuel in the presence of zeolite catalysts (HZSM-5, Amberlyst-70) [2].

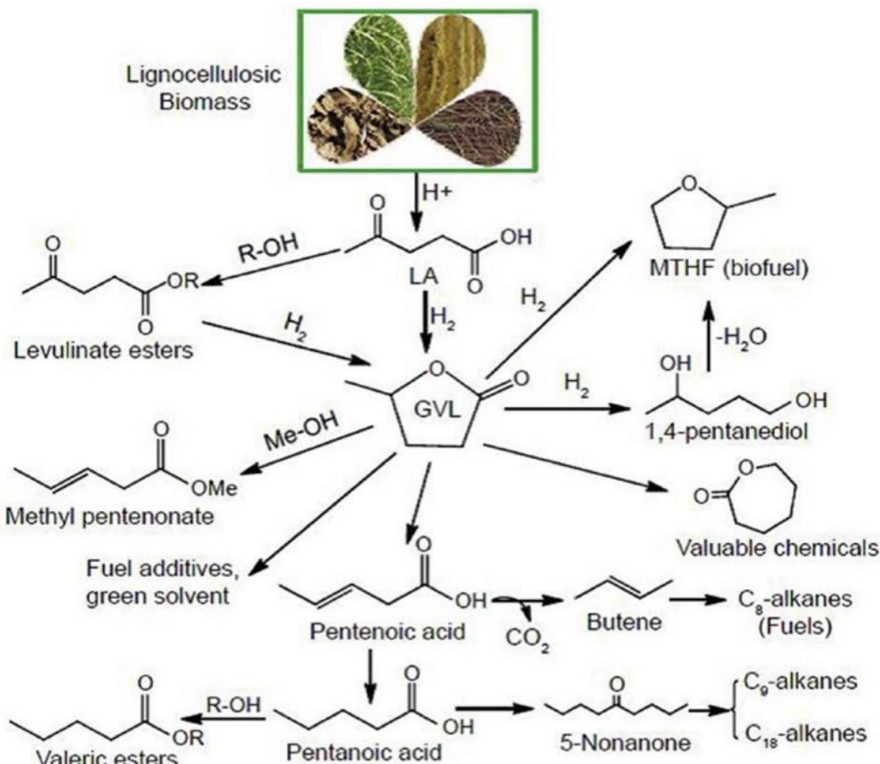

**Figure 3.** Schematic illustration of different important products that can be derived from GVL. Reproduced from ref [35] with permission from Elsevier.

Catalytic conversion of LA to GVL and other value added chemicals (such as those shown in Figures 2 and 3) using supported noble (Ru, Pt, Pd, Au) [33,36–40] and non-noble (Ni, Mo) [41] metal catalysts has been reported in the past two decades. For example, Jiang et al. reported MgO-Al$_2$O$_3$ supported Ni catalyzed hydrogenation of LA to GVL [40]. Other studies on supported Ni catalyzed hydrogenation of LA to GVL have been reported by other authors [29,42,43]. Of these reports, it has been previously shown that Ru catalysts are more active than other noble and non-noble metals in the aqueous phase hydrogenation of LA [33,44]. However, in the gas phase hydrogenation of ketones, Ru catalysts were observed to be less active than other noble metals [45]. Some explanation on why Ru-based catalysts have tend to display excellent activity for the aqueous phase hydrogenation of carbonyl compounds has been reported by other Michel et al. [46]. To date, there are several review articles on the catalytic hydrogenation of LA to GVL. All these reviews focus on either noble metal [33,47–49] or non-noble metals [50] as catalysts. Of all these review articles, only one recently published review by Seretis et al. [48] has focused on the heterogeneous and homogeneous Ru-based catalysts for this hydrogenation of LA to GVL. In their review, a wide range of heterogenous Ru-based catalysts were discussed, especially TiO$_2$ and carbon supported catalysts. Another recent review by Adeleye et al. [47] did

not extensively focus on the transformation of LA to GVL, but rather on the synthesis of other products derived from LA such as 2-butanol, 2-pentanol, and MTHF. In addition to systems discussed in those published reviews, this current review has discussed the effect of support and solvent on the catalytic performance and selectivity.

Additionally, a novel study on impurities in the LA feed on the catalyst's performance and stability for continuous-flow systems is also discussed. Additionally, the confinement of heterogeneous Ru catalysts with ionic liquids for easy product separation is also discussed in this review. Although there are several reports on the Ru based homogeneous catalyzed hydrogenation of LA to GVL [3,13,51–53], the discussion in this review will solely be devoted to supported Ru (including Ru containing bimetallic catalysts) catalyzed hydrogenation of LA to GVL. This is primarily because heterogeneous catalysts can easily be separated from reaction products and be re-used, making them ideal "green catalysts" as compared to their homogeneous counterparts. Other factors such as catalyst support (TiO$_2$, SiO$_2$, Al$_2$O$_3$, C, etc.) and the effect of solvent on the activity, selectivity, and product yield will also be discussed. It should be highlighted that ruthenium has recently been listed among other critical metals [54]. It has been reported that the recovery of ruthenium from process catalysts was not common until recently. This is because costs associated with the recovery processes may exceed the value of the ruthenium recovered since ruthenium is less costly.

## 2. Supported-Ru Catalyzed Conversion of LA to GVL

### 2.1. Catalytic Hydrogenation of LA to GVL over Differently Supported Ru Catalysts

As highlighted in the introduction section, the catalytic hydrogenation of AL has been performed using Ru-based catalysts immobilized on various support materials as a way of developing stable and greener catalysts. Of these supports materials, carbon and titania appear to have been the most reported as Ru supports for the LA hydrogenation reaction. However, other conventional supports such as silica, ZrO$_2$ and zeolites have also been used for the fabrication of active, stable, Ru-based heterogeneous catalysts. In some some instances, stable, highly active, supported Ru catalysts based on novel supports (such as hydroxypetite and ionic liquids) have been reported for this particular reaction. The choice of support has been found to has an effect on the selectivity of product and catalyst stability. In the coming sections, reports on supported-Ru catalyzed hydrogenation of LA to GVL are discussed with the aim to analyze the effect of support, solvent, and particle size on the activity, selectivity, and stability of the catalysts.

### 2.2. Carbon-Supported Ru Catalyzed Hydrogenation of LA

As briefly highlighted in the introduction section, carbon-supported Ru catalysts have been perceived as the most active catalysts reported in the hydrogenation of LA to GVL thus far. However, some of the limitations of Ru/C catalysts have been that they are not stable because they easily aggregate or leach out of the carbon support materials during catalytic reactions. For example, Yan and Liu published a study on the hydrogenation of LA to GVL over Ru/C (5% metal loading) catalysts [55]. For comparison purposes, Pd/C, Raney nickel, and Urushibara nickel catalysts with the same metal loading as that of Ru were also screened for the same reaction. The Ru/C catalyst was found to show higher activity as compared to other catalysts. This higher activity of Ru/C was attributed to small particle size as well as good metal dispersion on the carbon support. After roughly 2.5 h, the conversion of LA as well the selectivity to GVL was observed to have reached a plateau. Similarly, at the pressure of 1.2 MPa, maximum conversion and selectivity were reached. Both LA conversion and selectivity towards GVL decreased by 50% after the fourth catalyst recycle. This was ascribed to significant leaching of Ru nanoparticles out of the carbon support during catalytic runs of the recyclability study as determined by X-ray photoelectron spectroscopy (XPS) analysis.

On the contrary, Galletti et al. demonstrated that Ru/C can remain stable even after 5 reaction cycles when used with a heterogeneous acid co-catalyst (such as niobium,

phosphate, or ion exchange resins Amberlyst) [56]. In their study, they used commercial Ru catalysts (5%) supported on carbon and alumina. The Ru/C (without co-catalyst) was generally observed to show higher activity compared to other studied catalysts with 100% LA conversion and activity of 569.6 h$^{-1}$. Heterogeneous acidic co-catalysts were found to enhance the performance of the Ru/C supported catalyst (see Figure 4). The ion exchange resin Amberlyst as co-catalyst showed the highest activity (555.2 h$^{-1}$) as compared to all other co-catalysts used. Two reaction mechanisms leading to the formation of GVL were proposed in this study. One path involves the formation of intermediate γ-hydroxyvaleric acid followed by the removal of water (esterification) to form GVL. The other second path entails the formation of the thermodynamically unstable enol form of levulinic acid, which undergoes esterification to form angelica lactone (AL) which is subsequently hydrogenated to give GVL (see Scheme 1). Manzer screened several carbon-supported catalysts (Ir, Rh, Pd, Ru, Pt, Re, Ni) in the hydrogenation of LA to GVL [57]. The Ru/C catalyst was found to be more active and gave higher selectivity towards GVL relative to other screened catalysts.

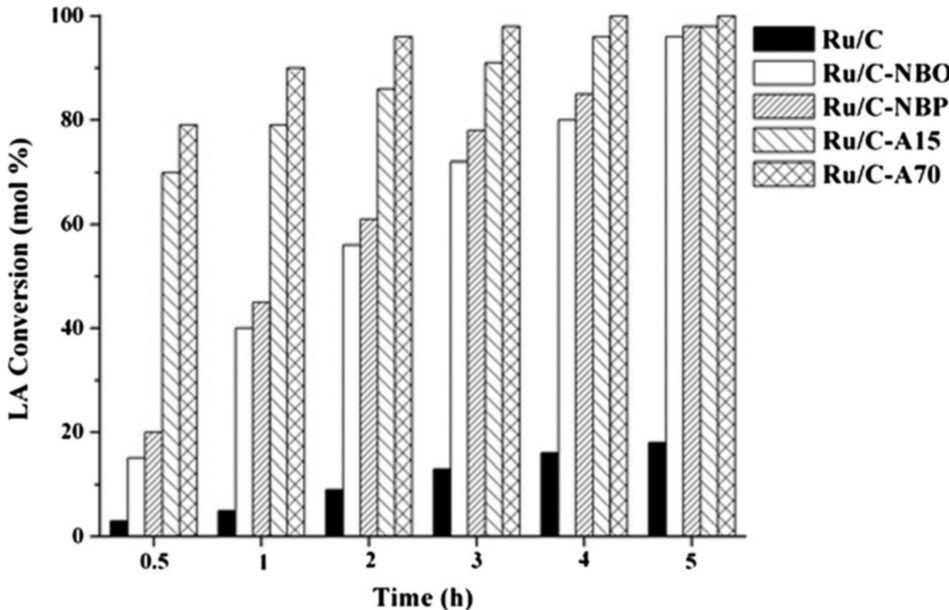

**Figure 4.** Effect of different heterogeneous acid co-catalysts on levulinic acid (LA) conversion (mol%) vs. time (h), employing 5% Ru/C under mild conditions. Reproduced from ref [56] with permission from The Royal Society of Chemistry.

**Scheme 1.** Proposed mechanism for the catalytic selective hydrogenation of LA to GVL, 1,4-pentanediol (PD), and methyl tetrahydrofuran (MTHF) [3].

Al-Shaal et al. studied the influence of the solvents (methanol, ethanol, 1-butanol, 1,4 dioxane, methanol-H$_2$O, ethanol-H$_2$O, butanol-H$_2$O, and H$_2$O) in the Ru-C catalyzed

hydrogenation of LA [58]. Other inorganic supports such as $TiO_2$, $Al_2O_3$, and $SiO_2$ were also investigated for comparison purposes. Among used alcohol solvents, methanol was found to show the highest LA conversion and GVL yields. This observation was attributed to the high solubility of $H_2$ in the solvent as compared to others used in this study. The lowest LA conversion (48.6%) was obtained when 1-butanol was used. A significant increase in the conversion of LA was observed when the $H_2$ pressure was increased from 12 bar to 20 bar for 1-butanol. However, the selectivity for 1-butanol was still comparable to what was observed for other solvents.

A significant enhancement in the conversion of LA was observed when a mixture of 1-butanol-water (90/10% by volume) was used as a solvent. This observation further confirmed that $H_2$ has a higher solubility in water than other used alcohol solvents. The highest LA conversion (99.5%) was generally obtained using water as a solvent. Surprisingly, no change in terms of LA conversion was observed when the methanol-water solvent was used. In an attempt to optimize reaction conditions, solvent free-reactions were also carried out. However, in this case, prolonged reaction time (50 h) was required to achieve 100% LA conversion under mild reaction conditions (25 °C). A complete conversion of LA was obtained within 40 min of reaction time albeit at elevated temperature (190 °C). In some instances, the intermediate (AL) that forms before GVL tends to polymerize over the acidic catalyst's surface. This phenomenon leads to the loss of carbon or complete deactivation of the catalyst.

In the same year, the cascade method for the conversion of cellulose to GVL was reported by Dumesic's group [1]. Their method involves the decomposition of cellulose via hydrolysis followed by subsequent dehydration of the resulting glucose over sulfuric acid as a catalyst to produce equimolar amounts of LA and FA in a batch reactor. The LA was then hydrogenated to GVL using FA as a source of $H_2$ in the presence of commercially obtained Ru/C catalyst. After 3 h of reaction time, both the LA conversion and GVL yields were observed to be above 90%. The produced GVL then separated from the aqueous sulfuric acid catalyst and was subsequently hydrogenated further to give other useful products such as pentanoic acid (PA) as an intermediate. The PA was then converted to 5-nonanone (a precursor to hydrocarbon fuels) using a niobium oxide-supported Pd catalyst.

Dumesic's group later reported an alternative method to avoid the deactivation of the catalyst for the conversion of LA to GVL, which was subsequently converted to 5-nonanone using Ru/C catalyst [24]. In this way, the LA hydrogenation formed, GVL, was subsequently hydrogenated further to give 5-nonanone at a lower temperature (150 °C). The used method and reaction conditions allowed for the reaction to favor the formation of 5-hydroxypentanoic acid (HAP), instead of AL, before the formation of GVL. The Ru/C catalyst showed more activity and selectivity for GVL as compared to other catalysts (Pd/$Nb_2O_5$, $CeZr_{0.5}O_2$) for the same reaction under similar reaction conditions. A full LA conversion and a selectivity towards GVL of 96% was obtained for Ru/C at ambient temperature.

The catalytic hydrogenation of LA to GVL over Ru/C and bimetallic RuRe/C catalysts has been reported by the same group as well [59]. They observed that both mono- and bimetallic catalysts showed more stability in the presence of sulfuric acid, a common catalyst used in the conversion of cellulose to LA. In this case, they started with the conversion of cellulose to LA and formic acid using $H_2SO_4$ as a catalyst in a batch reactor. Both monometallic Ru/C and RuRe/C catalysts were prepared using the conventional incipient wetness impregnation method. Formic acid was used as a source of hydrogen in a flow reactor. The initial rates of the reaction were observed to be high at the beginning of the reaction for the Ru/C catalyst and started decreasing over time. After 50 h, sulfuric acid was introduced into the reaction feed and the catalyst started to show more stability. However, after the introduction of sulfuric acid, the selectivity towards GVL was observed to decrease from >98% to 60–70%. This decrease in selectivity was attributed to the formation of LA-sulfuric acid anhydride species that formed upon the addition of sulfuric acid. The formation of this anhydride has been reported to be reversible and catalyst

independent [60]. On the contrary, however, no effect on the selectivity for GVL in the presence or the absence of the sulfuric acid in the case of RuRe/C catalyst was observed. Instead, an increase in the activity by a factor of two was observed in the presence of sulfuric acid as compared to the Ru/C catalyst. This was due to the enhanced activity of the bimetallic RuRe/C catalyst relative to Ru/C counterpart.

In another study, Domesic's group reported a comparative study between the monometallic Ru/C and the bimetallic RuSn/C catalysts in the selective hydrogenation of LA to GVL [61]. Commercial monometallic 5 wt% Ru/C was used and bimetallic ($Ru_xSn_y$/C) catalysts were prepared via incipient wetness impregnation. The catalytic hydrogenation was carried using 2-sec-butyl-phenol (SBP) as a solvent. Surprisingly, the Ru/C catalyst was found to exhibit more activity than the bimetallic RuSn/C catalyst. However, the Ru/C showed less stability overtime on stream compared to the RuSn/C, which maintained its initial activity even after 300 h of time on stream. Of the various RuSn/C catalysts with different metal ratios, the highest stability was displayed by the catalyst with higher Sn concentration. The choice for this solvent (SBP) was based on its ability to easily extract LA from sulfuric acid which is used as a catalyst during LA production from cellulose. This was after a previous study from the same group showed that alkylphenol solvents can successfully extract LA from aqueous solutions of sulfuric acid [62]. This method, however, has limitations in that: (i) it requires final purification of the product by distillation, which can be costly, (ii) the partition coefficient of LA extraction is relatively low and (iii) the partition coefficient of the extracted FA was less than 0.2, making it not possible to use it as an internal source of $H_2$ for hydrogenation reaction.

In trying to overcome such limitations, the same research group published some work in which GVL was used as a solvent to extract both LA and FA from aqueous solutions [63]. The use of GVL as a solvent during the production of LA and FA from cellulose materials enables for easy separation/extraction of LA and FA into the organic phase. In the case where the targeted product is GVL for the hydrogenation of LA, the need to separate the product from the solvent was eliminated. Additional advantages of using GVL as a solvent in the cellulose decomposition process to produce LA and FA is that GVL solubilizes humin formed during the process. The conversion of LA to GVL was carried out in the presence of commercially obtained Ru-Sn/C (5 wt%) catalyst. They found that the conversion rate of cellulosic LA was lower as compared to that of the commercial LA under similar reaction conditions. This was attributed to the impurities generated during the decomposition of cellulose materials to produce LA and FA. The catalyst was, however, observed to show stability even after 40 h in the reaction feed. This indicated that the "assumed" impurities do not necessarily deactivate the Ru-Sn/C catalyst. It is also noteworthy to mention that the LA conversion was found to increase as the temperature is elevated without hydrogenation products of the GVL solvent forming.

Yang et al. also recently reported a comparative study on the use of monometallic Ru and bimetallic Ru-Ni nanoparticles supported on ordered mesoporous carbon (OMC) as catalysts in the hydrogenation of LA to GVL [64]. The synthesis of OMC-supported Ru or Ru-Ni nanoparticles was carried out using a novel procedure depicted in Figure 5. The metal precursors were coordinated with the modified chitosan to form the precursor composite materials by cetyl trimethylammonium bromide (CTAB)-directed self-assembly. This resulted in the formation of bimetallic Ru-Ni nanoparticles embedded on the mesoporous carbonaceous framework, as shown in Figure 5. The nanoparticles were formed during pyrolysis at high temperature (750 °C) without any additional reductant and the silica was removed by etching with an aqueous solution of concentrated NaOH (3.0 M). All OMC-supported Ru and Ru-Ni catalysts showed excellent activity for the hydrogenation of LA and showed high selectivity towards GVL. The bimetallic OMC-supported Ru-Ni catalyst exhibited more active sites than the monometallic Ru counterparts under similar reaction conditions. For instance, a turn over frequency for the Ru-OMC catalyst was calculated to be 1716 h$^{-1}$ as compared to 870 h$^{-1}$ determined for commercial Ru/C catalyst. The Ru-Ni-C catalyst prepared via direct pyrolysis of chitosan-Ru-Ni coordination

showed the lowest activity, giving LA conversion and GVL yield of 34 and 15%, respectively. This suggested that the textural properties of the support material play a crucial role in the overall performance of the catalyst. Similarly, the catalyst with unetched silica (Ru-Ni-OMSC) showed appreciable activity, however, lower than that observed for silica-free Ru-Ni-OMC catalysts. Generally, the bimetallic Ru-Ni-OMC catalyst showed higher activity compared to the Ru-OMC catalyst when subjected to similar reaction conditions. The bimetallic catalyst with the highest Ru content ($Ru_{0.9}Ni_{0.1}$-OMC) was found to exhibit more activity and stability than all monometallic and bimetallic catalysts. This catalyst was also chosen as a representative catalyst for the investigation of water as a potential solvent. Under similar reaction conditions as in the absence of a solvent, similar LA conversions were observed, however, the GVL yield was found to have slightly decreased from 97% to 84%. The $Ru_{0.9}Ni_{0.1}$-OMC catalyst was observed to maintain its activity after being used in 15 reaction cycles. This was attributed to higher pore volume, high surface area, and high Ru dispersion on the mesoporous carbon support.

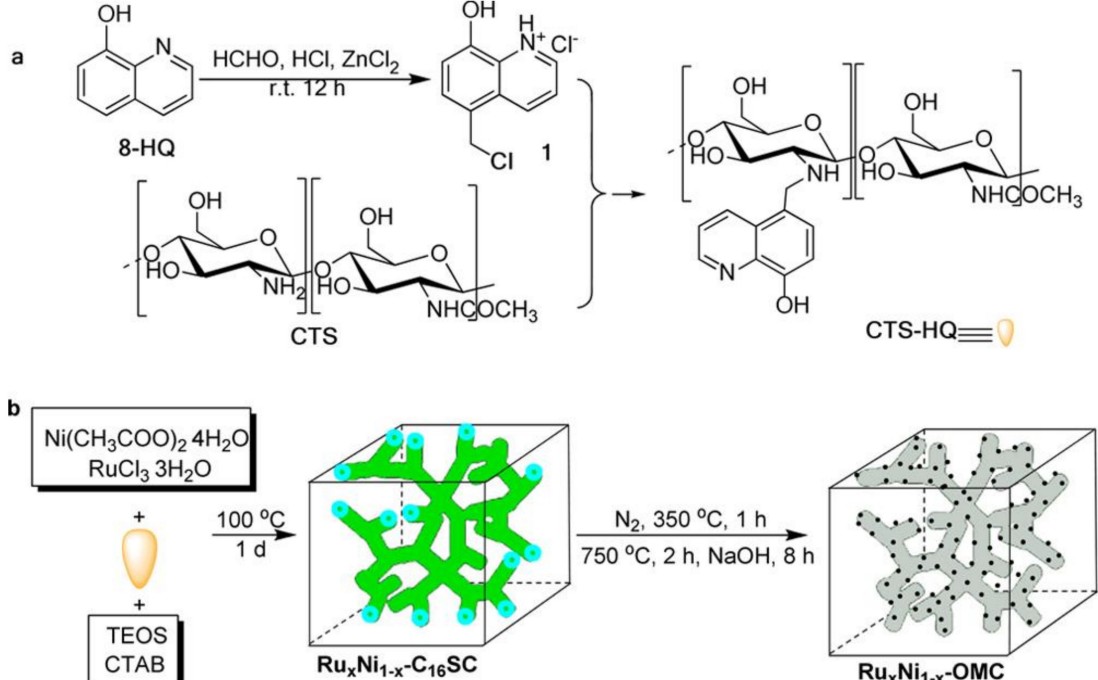

**Figure 5.** Synthesis of $Ru_xNi_{x-1}$-ordered mesoporous carbon (OMC) catalysts, (**a**) Modification of chitosan with 5-chloromethyl-8-quinolinol; (**b**) Synthetic route for the preparation of OMC-supported bimetallic catalyst from precursor composite ($Ru_xNi_{1-x}$-$C_{16}SC$) and modified chitosan. Reproduced from ref [64] with permission from American Chemical Society.

Rhenium has also been used to promote the activity of carbon-supported Ru catalysts for the hydrogenation of LA [65]. Monometallic and Re promoted carbon-supported Pt and Pd were also evaluated under similar reaction conditions as those applied for Ru-based catalysts. The monometallic 2.1% Ru/C was observed to show superior activity for the conversion of LA to GVL. About 100% LA conversion was achieved for 2.1% Ru/C within just 10 min of reaction time, while under similar conditions, 70% and 53% LA conversion were observed for 2.7% Pd/C and 4.1% Pt/C catalysts after 2 h, respectively. The bimetallic ReRu/C catalyst also showed similar if not slightly higher activity compared to the monometallic Ru catalyst counterpart. It is noteworthy to mention that, in the case of Re promoted Pd and Pt, a significant improvement in activity was observed. For instance, complete LA conversion was achieved for Pd−Re/C and Pt−Re/C catalysts after 2 and 7 h, respectively. However, the bimetallic 2.8% Ru−3.9% Re/C catalyst gave the highest GVL yield (85%). The formed product, GVL, was further hydrogenated to PD. For example,

in the case of the 2.8% Ru−3.9% Re/C, complete GVL conversion was achieved after 24 h with 55% selectivity towards PD.

Upare et al. published a study on the production of GVL from LA under vapor phase conditions using commercially purchased carbon-supported Ru catalyst (5 wt%) [66]. The reactions were run on a continuous-flow fixed-bed reactor system. For comparison, other carbon supported metal catalysts (5 wt% Pd or Pt) were also studied for the same reaction conditions. All studied catalysts gave 100% LA conversion. However, the highest selectivity for GVL was observed for the Ru/C catalyst (98.6%), while 90% and 30% GVL selectivity were obtained for Pd and Pt catalysts, respectively. The effect of pressure (from 1–25 bars) on the activity and selectivity was also investigated in this study. It was found that the activity and selectivity were not affected when the pressure was varied from 1–5 bar for Ru/C catalyst. A further increase to the pressure led to an increase in the selectivity of other side products such as MTHF and AL while the selectivity for GVL marginally decreased. Generally, the Ru/C showed no lowered activity for 10 days in the continuous flow reactor.

On the contrary, Bond and co-workers observed different product selectivity for the Ru/C (commercial, 5 wt%) catalyzed hydrogenation of LA at low temperatures [67]. In their work, two possible reaction pathways were proposed leading to the formation of GVL. One path involves the hydrogenation of LA 4-hydroxypentanoic acid, which subsequently undergoes esterification to form GVL. The other path entails the formation of AL via dehydration of LA before the formation of preferred GVL. This is in agreement with what was proposed by other researchers [56]. It was observed that when the reaction temperature is near ambient (50 °C), the selectivity for GVL was very poor (around 5%) as compared to that of HAP (<95%). However, in the absence of $H_2$ under an inert atmosphere ($N_2$), the formation of AL was found to dominate over HAP. It was concluded from these observations that in the absence of $H_2$, LA is solely consumed by dehydration. Secondly, the GVL is formed through the formation of HAP during the Ru/C catalyzed hydrogenation of LA. Furthermore, the following conclusions were drawn after carrying out a thorough kinetics study: (i) at a lower temperature, the esterification of the intermediate (HAP) tends to control the rate of GVL formation, (ii) at elevated temperature, mass transfer limits the rate of LA hydrogenation, (iii) the production of GVL can potentially be enhanced by using a combination of a strong acidic catalyst and hydrogenation metal catalysts, (iv) a deactivation of Ru/C overtime on stream was observed and the causes could not be established at the time of the publication. Recently, Piskun et al. reported the hydrogenation of LA using carbon-supported Ru particles as catalysts [44]. Unlike in other studies where the sizes of the Ru particles are usually in the nanometer range, the Ru particles used for this study were in the millimeter range. Ru particles in the range of 1.25–2.5 mm were supported (0.5–2 wt%) on other inorganic supports such as $TiO_2$ and $Al_2O_3$. However, only the carbon-supported Ru catalyst will be discussed in this section.

The carbon-supported Ru catalyst was observed to exhibit excellent activity compared to those supported on different oxide supports. After 6 h of time on stream, about 92% of LA conversion was observed. The selectivity towards GVL was found to be 77% under reaction conditions employed (T = 90 °C, Pressure = 45 bar). The effect of the pre-reduced catalyst on the activity was also investigated using the same catalyst. In that way, the Ru/C catalyst was pre-reduced inside the reactor at 350 °C for 4 h under the flow of a mixture of $H_2$ and $N_2$ gas. It was found that this pre-reduction step of the catalyst results in a decrease in the activity. For instance, the conversion of LA observed for a reduced catalyst was 75% compared to 95% for unreduced catalyst under similar reaction conditions. This observation was attributed to an increase in the size of Ru particles during the catalyst activation process. The stability of the Ru/C was also probed at 130 and 150 °C, respectively. A notable decrease in the catalyst's activity with time was observed for a reaction run at 150 °C as opposed to the one run at 130 °C. Very recently, We et al. reported the synthesis and catalytic investigation of N-doped hierarchically porous carbon-supported Ru catalyst (Ru/NHPC) in the hydrogenation of LA to GVL [6]. The N atom was doped

to the hierarchically porous carbon (NHPC) to improve the interaction between carbon and Ru nanoparticles and subsequently enhance the catalyst stability. This was to try to circumvent the deactivation of the traditional Ru/C catalysts that often suffer from loss of activity during the reaction, even under mild reaction conditions. Moreover, activated porous carbon is believed to depress the mass transport of molecules, which may result in the blocking of pores [68]. The average size of the NHPC supported Ru nanoparticles synthesized via the ultrasonic-assisted deposition method was determined to be 2.1 nm.

The different parameters such as temperature, solvent, and pressure were investigated in the catalytic hydrogenation of LA to GVL. Of all the solvents (cyclohexane, 1,4-dioxane, DMF, ethanol, ethanol, and water) investigated, water gave the highest LA conversion (>99%) even at the lowest temperature (25 °C). Although 1,4-dioxane and DMF gave almost 100% LA conversion, their selectivity towards the anticipated product, GVL, was very poor (1%). The effect of temperature on the activity and selectivity towards GVL was investigated employing water as a solvent. It was observed that no significant change occurred on the conversion of LA. However, the selectivity towards GVL was observed to increase as the temperature is elevated. For instance, GVL selectivity increased from 77% to >99% when the temperature is raised from 25 to 50 °C. The Ru/NHPC was also compared with a commercial Ru/AC (AC = activated carbon) as well as other commercial catalysts such as Ru/C, Pd/C, and Pt/C. Surprisingly, the Ru/NHPC was found to show excellent activity compared to all other catalysts studied, including Ru/AC, under similar reaction conditions. This was attributed to the presence of N atoms which improved the hydrophilic nature of the Ru/NHPC catalyst. As a result, the Ru/NHPC catalyst was found to have a good dispersion in water as compared to Ru/AC, thereby enhancing the activity. Interestingly, the Ru/NHPC was found to retain its activity even after being aged in the air for 1 year. Moreover, the catalyst maintained its activity even after 13 reaction cycles without significant loss of activity.

In some instances, commercial Ru/C catalysts often suffer from deactivation in the hydrogenation of LA to GVL in either batch or continuous systems. For example, some authors have reported deactivation of Ru/C in MeOH solvent just after the first catalyst recycle run, even at moderate temperature (130 °C) [55]. In trying to circumvent such limitations for Ru/C, Cao et al. recently prepared $Ru/ZrO_2@C$ catalyst with excellent stability [69]. The $Ru/ZrO_2@C$ was prepared via wet impregnation and well-dispersed Ru particles with an average size of 3.3 nm were obtained.

*2.3. Titania-Supported Ru Catalyzed Hydrogenation of LA*

Primo et al. have reported the study on the synergetic effect between the supports and the nanoparticles in the hydrogenation of LA [70]. They reported that $TiO_2$ supported Ru nanoparticle catalyst is 3 times more active for hydrogenation reactions (LA hydrogenation included) compared to the conventional carbon-supported Ru counterparts. In addition to LA, other substrates such as succinic acid and itaconic acid were screened to evaluate the effect of support on the hydrogenation rate. To arrive at their conclusion, they normalized Ru nanoparticles sizes on other supports such as C and $CeO_2$ for comparison purposes and found that the TOF for the $TiO_2$ supported catalyst was much higher than for Ru supported on other supports, for a Ru loading of less than 2%. This enhanced activity was attributed to the good dispersion of Ru nanoparticles on the titania support. Moreover, this enhanced activity for titania supported Ru catalysts was attributed to the synergetic effect between the titania support and Ru nanoparticles which subsequently activate the carbonyl group on the $TiO_2$.

The role of water in the metal-catalyzed hydrogenation of ketones (including LA) was also further investigated using both experimental and theoretical studies by Michel et al. [36]. In their study, titania supported Ru, Pd, and Pt catalysts were evaluated in the hydrogenation of LA to GVL under two different environments (water and THF) and mild conditions (70 °C, 50 bar of $H_2$). The catalysts for all metals are prepared in such a way that reproducible average sizes can be obtained to minimize the effect of particle sizes on

the catalyst's performance. The average sizes of these metal particles were determined by TEM analysis in the range of 2.1–3.2 nm. Additionally, the active metal sites were observed to be homogeneously well-dispersed on the titania support. As a result, that these metal catalysts were all supported on the same support (titania), it was possible to conduct a comparative study on the effect of two solvents in the hydrogenation of LA to GVL under identical reaction conditions.

Both density functional theory (DFT) and experimental data were used to study the role of these two solvents. Both experimental and theoretical results revealed that Ru has a better activity for LA hydrogenation, with the highest activity observed in the $H_2O$ solvent. However, the same was not the case when the Ru catalyst was used in the THF solvent as no activity was observed. This enhanced activity of Ru in the $H_2O$ solvent was attributed to the hydrogen bond effect of water, which reduces the reaction's energetic span.

A related study on this enhancement in the activity of the Ru supported catalysts in aqueous medium was also reported by Tan et al. [71]. In their study, $TiO_2$, $SiO_2$, and $ZrO_2$ were used as supports for the Ru catalyst and compared in the hydrogenation of LA to GVL. Titanium supported Ru catalysts showed the highest activity compared to other screened catalysts in aqueous media and moderate temperature (70 °C). The $TiO_2$ supported catalysts with low Ru loading were generally observed to exhibit superior activity for LA hydrogenation. Good dispersion of Ru nanoparticles on the $TiO_2$ support was cited as the reason for this excellent activity. The investigation on the effect of solvent on the activity of the titania supported Ru catalyst was also carried out. Of all solvents screened ($H_2O$, ethanol, ethanol, and 1,4-dioxane), water was found to give high activity while maintaining high selectivity for GVL. Since $H_2$ has very low solubility in water, this promotional effect was attributed to the ability of water to promote the distribution of H atoms on the catalyst's surface. Additionally, LA conversion was found to improve as the amount of water solvent was increased. The possibility of water participating in the reaction was suggested for this observation. At higher Ru loading (2 wt%) and elevated temperature (130 °C), all catalysts screened achieved the complete conversion of LA in just 30 min. However, when Ru loading is decreased (to 0.5 wt%), the conversion of LA decreased by almost 20% even after a reaction time of 3 h. Surprisingly, at moderate loading (1 wt%) and temperature (70 °C), the activity of the catalysts improved significantly with $TiO_2$/Ru giving 100% conversion after 6 h. This enhanced activity of the $TiO_2$/Ru catalyst with low metal loading was also attributed to the good dispersion of the Ru particles on the $TiO_2$ surface. It is noteworthy to mention that some authors have reported a kinetic and mechanistic study on the Ni/$TiO_2$ catalyzed hydrogenation of LA. They found that Lewis acid sites on the $TiO_2$ surface contribute towards enhancing the selectivity for the formation of GVL [43].

Luo et al. also reported a study on the enhanced activity of titanium supported bimetallic Ru-based catalysts in the catalyzed hydrogenation of LA to GVL when alloyed with Pd nanoparticles [37]. These catalysts were synthesized using a modified impregnation method for controlled particle sizes with a narrow size distribution. It should be noted that this modified impregnation method involves the addition of excess chlorine (in the form of dilute HCl) to achieve particles with smaller sizes. Encouraged by an observed tremendous improvement in the activity of the Au-Pd/$TiO_2$ in the hydrogenation of LA to GVL, they prepared an alloy of Ru-Pd/$TiO_2$ and compared (catalytic activity) it with the monometallic Ru/$TiO_2$ counterpart. It was found that the bimetallic Ru-Pd/$TiO_2$ gave almost complete conversion of LA after 30 min, while a full LA conversion was observed after 40 min for the monometallic Ru/$TiO_2$ catalyst. At longer reaction times (>2 h), the selectivity for GVL was observed to decrease for Ru/$TiO_2$, while the bimetallic Ru-Pd/$TiO_2$ remained stable. This stability observed for the bimetallic Ru-Pd/$TiO_2$ catalyst was attributed to the presence of the Pd species that diluted and isolated Ru sites during the reaction. To understand the effect of excess HCl on the performance of the catalyst, monometallic Ru/$TiO_2$ was prepared (without adding excess HCl). This catalyst was observed to be superior to both the monometallic Ru/$TiO_2$ and bimetallic Ru/$TiO_2$ prepared in the presence of dilute

HCl. However, less sintering was observed for monometallic $Ru/TiO_2$ as compared to the bimetallic $Ru-Pd/TiO_2$ catalyst.

The influence of support and solvent on the catalyst selectivity and stability in the hydrogenation of LA to GVL has also been reported by Lu et al. [72]. $TiO_2$, $Nb_2O_5$, H-β, and H-ZSM5 were all used as supports for Ru nanocatalysts (1 wt% loading) via wetness impregnation. The influence of the solvent on the hydrogenation of LA was investigated using 2-ethyl hexanoic acid (EHA) and dioxane solvents. The zeolites supported catalysts (Ru/ZSM5 and Ru/H-β) were found to be more acidic than the $Ru/TiO_2$ and $Nb_2O_5$ catalysts, with $Ru/Nb_2O_5$ being the least acidic. Consequently, hydrogenation products of GVL (pentanoic acid and ester derivatives were observed in a significant amount when the more acidic catalysts were used in dioxane as the solvent. The formation of ester derivatives was attributed to the esterification of pentanoic acid with dioxane decomposition products. The decomposition of dioxane was caused by its instability in the presence of acidic catalysts under the reaction conditions employed. Indeed, this observation proved that LA can potentially be directly converted to PA in a one-pot reaction. However, it is worth mentioning that the metal/zeolite catalyzed hydrogenation of GVL to PA has already been patented by a petrochemical company, Shell [73].

The extent of catalyst stability in the dioxane solvent was also investigated during the course of the reaction. After 4 h, the Ru/H-ZSM5 catalyst was observed to have lost about 2.4% of Ru metal. The Ru nanoparticles size increased from 3.5 nm to 4 nm before and after the reaction, respectively. Post-characterization by powder XRD revealed that the support materials did not undergo any structural/textural change during the reaction. However, $N_2$-physisorption for spent catalysts showed that there was a decrease in surface area and pore volume for $Nb_2O_5$ and zeolite supported catalysts. To try and circumvent problems encountered when dioxane was used as a solvent, EHA was used as an alternative solvent as it is stable under hydrogenation reaction conditions. A significant difference in terms of activity and selectivity was observed when the EHA solvent was used under similar conditions as applied for dioxane. For instance, the selectivity for GVL significantly improved for $Ru/TiO_2$ and $Ru/Nb_2O_5$ catalysts. Full LA conversion was observed for the $Ru/TiO_2$ catalyst after 4 h, while only 2.4 mol% of LA was still unconverted for the $Ru/Nb_2O_5$ after 5 h. Similarly, the Ru/H-β catalyst also gave a full LA conversion after 4 h, whereas 100% conversion of LA was observed after 3 h for the Ru-ZSM5 catalyst. Only GVL was observed as the product for $Ru/TiO_2$ and $Ru/Nb_2O_5$ catalysts. This was mainly because these catalysts were rendered to be weakly acidic catalysts and as a consequence, the ring-opening of GVL was not possible under conditions employed. As expected with more acidic catalysts, a significant amount of PA was observed after a reaction time of 10 h. The amount of PA observed in the EHA solvent was lower than that produced in the dioxane solvent. This was attributed to the competition that exists among LA, GVL, and solvent for the adsorption or interaction with the catalyst.

At low LA conversion, the Ru/H-β catalyst was found to produce a significant amount of PA. This was attributed to more available acidic sites accessible (responsible for the formation for GVL ring-opening) by GVL in the Ru/H-β catalyst as compared to the Ru/ZSM5 counterpart. However, after a lengthy reaction rime (10 h), the selectivity of PA for the Ru/H-β catalyst was found to be lower as compared to that for Ru/ZSM5. This suggested that the Ru/H-β catalyst is more prone to deactivation than Ru/ZSM5 catalyst. The catalyst stability study in the EHA solvent showed comparable results to what was observed in the dioxane solvent except for the Ru/ZSM5 catalyst, that suffered severe sintering and particle agglomeration (confirmed by TEM and EDX).

Solvent-free reactions were also performed to investigate the influence of acidity of the support on the activity and selectivity. For this, only Ru/H-β and $Ru/TiO_2$ catalysts were chosen. It was found that the catalysts showed more activity as compared when reactions are run in EHA solvent. For instance, the turn over frequency calculated for $Ru/TiO_2$ was $0.239\ s^{-1}$ compared to $0.164\ s^{-1}$ for Ru/H-β determined in the EAH solvent. Similarly, TOF for Ru/H-β increased from 0.131 to $0.403\ s^{-1}$ for reactions run in EAH and neat LA,

respectively. Indeed, this observation further confirms the completion between LA and EAH solvent for adsorption on the catalyst's surface. The stability of catalysts was also studied using neat LA with no solvent. It was observed that the Ru metal loss and sintering were relatively limited for Ru/TiO$_2$ compared to when there is solvent involved in the reaction. More coke formation was observed for the Ru/H-β catalyst than when the EHA solvent was used. Generally, more coke formation was observed for all catalysts compared to in EHA solvent, hinting that this coke formation is not induced by the presence of EHA solvent but rather by LA derivatives.

The influence of titania support on the GVL yield during Ru catalyzed LA hydrogenation was investigated by Ruppert and co-workers [74]. In their study, titania supports (high and low surface area) with different crystalline phases (anatase and rutile) were used as supports for Ru nanoparticles. To explore further, a catalyst consisting of a mixture of both rutile and anatase phase was also evaluated in the same reaction. These Ru-based, titania supported, catalysts were prepared by either incipient wet impregnation or liquid phase direct chemical reduction. The average particle sizes of Ru were determined by TEM to range between 2.1 nm to 4.5 nm. For all systems studied, all Ru-based catalysts were observed to exhibit superior activity in the hydrogenation of LA to GVL in terms of both conversion and GVL selectivity as compared to Pt-based catalysts screened for the same reaction. More interesting to note was that these titania supported Ru catalysts were found to give higher GVL yields than their carbon-supported counterparts.

For both reaction temperatures studied (30 °C and 70 °C), the anatase-rutile mixed-phase supported Ru catalysts showed the highest activity as well as high GVL yields. For instance, Ru/TiO$_2$ (rutile: anatase = 10:80%) gave 100% for LA conversion as well as GVL yield. On the other hand, with rutile: anatase ratio of 20:80% gave LA conversion of 99% and GVL yield of 95%. It should be mentioned, however, that even the Ru catalyst supported on pure rutile phase also gave appreciable results, giving 95% and 83% for LA conversion and GVL yield, respectively. The Ru catalysts supported on high surface area pure anatase phase showed relatively low activity. For example, the highest LA conversion and GVL yield obtained for these pure anatase phases supported Ru catalysts was 38% and 31%, respectively. The presence of micropores on the anatase phase support was cited as the reason for this reduction in activity as compared to either pure rutile or rutile-anatase mixed-phase supported Ru catalyst. Additionally, these micropores were observed to induce the formation of Ru agglomerates on the support's surface, thereby compromising the catalyst activity. To improve the activity of this pure anatase supported Ru catalysts, further calcination was performed (at 500 °C for 3 h). Indeed, a significant improvement in the activity was observed with the calcined catalysts, giving LA conversion and GVL yield of 99% and 93%, respectively. These values were comparable to those obtained for pure rutile supported or mixed phases supported Ru catalysts under similar reaction conditions. Two important key factors were cited for the difference in the activity of these different supports with different physicochemical properties: (i) electronic properties which facilitate the dispersion of particles on the oxide surface and (ii) the morphology of the oxide support which in turn affects the surface area and porosity and consequently the overall catalyst's activity. For instance, HRTEM analysis revealed that mixed oxide supported Ru nanoparticles prefers to grow on the rutile grains than on the anatase phase of the mixed-phase support. As a result, Ru catalyst supported on pure rutile phase showed significantly higher activity than those supported on pure anatase phase.

We have recently published work on the use of dendrimer-derived supported Ru nanoparticles in the hydrogenation of LA to GVL [75]. The dendrimer-derived Ru nanoparticles were immobilized on mesoporous TiO$_2$ support via wet impregnation and were denoted as Ru$_{40}$@Meso-TiO$_2$. The dendrimer template was removed by calcination at high temperature (550 °C) before catalytic evaluation of the supported Ru catalysts. The average sizes of the mesoporous titania supported Ru nanoparticles were determined by HRTEM to be 1.4 ± 0.2 nm (see Figure 6). The resulting Ru$_{40}$Meso-TO$_2$ catalyst was revealed by

TEM to be composed of the anatase phase of titania with small Ru particles well-dispersed within the support channels (Figure 6).

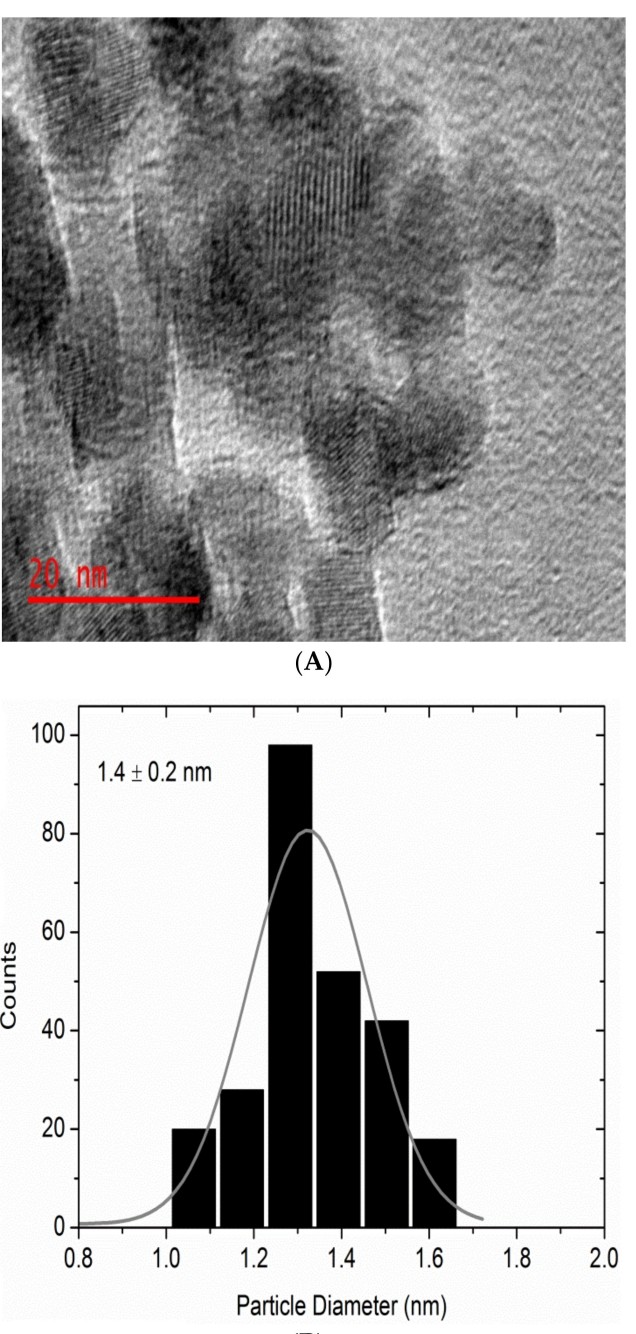

**Figure 6.** (**A**) HRTEM images of mesoporous titania supported Ru particles (Ru$_{40}$@Meso-TiO$_2$) and (**B**) the corresponding particle size distribution histogram for Ru$_{40}$@Meso-TiO$_2$. Reproduced from ref [75] with permission from Elsevier.

At a temperature of 150 °C, H$_2$ pressure of 10 bar, and reaction time of 5 h, both these catalysts were found to perform at their optimum level for the hydrogenation of LA to GVL. For instance, 98% LA conversion was observed for Ru$_{40}$@Meso-TiO$_2$. The Ru$_{40}$@Meso-TiO$_2$ catalyst showed higher selectivity towards the formation of GVL under conditions employed. Higher selectivity for GVL was attributed to the presence of Lewis acid sites on the titania support. The effect of solvent on the activity and selectivity of these mesoporous supported Ru catalysts was investigated using 1,4-dioxane and H$_2$O as solvents. In our

case, both solvents produced comparable results. However, some authors have reported an enhanced activity for $H_2O$ as compared to other solvents [71]. The $Ru_{40}$@Meso-$TiO_2$ catalyst proved to be robust even after 3 reaction cycles, thereby indicating that there was no severe leaching under reaction conditions employed.

Anatase $TiO_2$ supported ultrasmall $Ru(OH)_x$ nano-clusters have been reported to show excellent catalytic activity in the hydrogenation of levulinic acid as well as its levulinate esters (ML) [76]. Another oxide support such as $Al_2O_3$, MgO, $CeO_2$, $ZrO_2$, HT (Mg-Al hydrotalcite) was also used to support $Ru(OH)_x$ nanocatalysts for comparison purpose. The catalysts were evaluated via catalytic transfer hydrogen (CTH) using 2-propanol (2-PrOH) as a source of hydrogen under mild conditions (90 °C) for a reaction time of 24 h. Unsupported $Ru(OH)_x$ was found to achieve ML conversion and GVL yields of 52% and 47%, respectively. When the same amount of $Ru(OH)_x$ is supported on high surface area anatase $TiO_2$, higher ML conversion was observed. Among all catalysts screened for these reactions, $TiO_2$ supported Ru (0.8 mol%) catalysts gave the highest ML conversions.

Titanium containing pure anatase or anatase/rutile phase catalysts gave the highest GVL yields as compared to their pure rutile phase counterparts. For instance, pure anatase containing catalyst gave a GVL yield of 76% as compared to that obtained for pure rutile supported Ru catalysts of 49%. The effect of Ru loading on the performance of the catalysts was also investigated using pure anatase and anatase/rutile titanium supports. It was found that the catalyst activity decreases as the Ru loading is increased (from 0.8 wt% to 4.2 wt%) on the anatase/rutile titania support. However, the opposite was observed for the pure anatase titania support. For example, 4% $Ru(OH)_x$/$TiO_2$(A) gave the highest ML conversion of 99% and GVL yield of 89%. Using the optimum catalyst (4% $Ru(OH)_x$/$TiO_2$(A)), various alcohols (MeOH, EtOH, 2-BuOH, 1-PrOH, and CyOH) were also evaluated as a source of $H_2$ for CTH reactions. It was found that primary alcohols such as MeOH, EtOH, and 1-PrOH did not act as effective $H_2$ donors for the CTH reactions. This was attributed to the difficulty of β-H elimination from primary alcohols [77]. About 80% LA conversion and 64% GVL yield were obtained when the 4 mol% Ru catalyst was used.

Recently, another work describing the catalytic hydrogen transfer (CHT) for the $TiO_2$ supported Ru catalyzed hydrogenation of LA and its derivative, methyl levulinate (ML) to GVL has been reported [78]. The average particle sizes of these Ru/$TiO_2$ catalysts with different metal loading (2–5 wt%) were determined by TEM to be around 5 nm. 2-Propanol was used as both solvent and source of hydrogen for this CHT hydrogenation of ML. Their solvent choice was inspired by previous reports that showed that 2-propanol acts as a good $H_2$ donor with higher conversion and product selectivity [77]. Optimization of the reaction conditions was performed using the 5% Ru/$TiO_2$ catalyst. A good ML conversion (98%) and GVL selectivity (71%) was observed at 150 °C and an ML concentration of 0.3 mol/L. When the temperature was raised to 200 °C, ML conversion and GVL selectivity increased to 99% and 98%, respectively. Product selectivity decreased drastically with an increased inlet flow-rate. When the inlet flow-rate was increased from 0.3 to 0.5 mL/min, the ML conversion and GVL selectivity decreased to 90% and 41%, respectively. Instead of GVL formation dominance, the product intermediate, 4-hydroxypentanoate was observed as the main product. Commercial 5% Ru/C and 5% Ru/$Al_2O_3$ catalysts were also evaluated for the same reaction for comparison purposes. It was found that these commercial catalysts gave lower activity and selectivity as compared to the as-synthesized 5%Ru/$TiO_2$ under similar reaction conditions. For instance, 5% Ru/C gave ML conversion and GVL selectivity of 83% and 52%, respectively. Likewise, 5% Ru/$Al_2O_3$ also showed a dramatic decrease in both ML conversion (31%) and GVL selectivity (97%).

Wojciechowska et al. recently published a work on the catalytic evaluation of Ca-modified titania supported Ru catalysts in the hydrogenation of LA to GVL [79]. Their Ru catalysts were prepared by immobilization of the Ru metal on the preformed $TiO_2$, Ca-modified $TiO_2$ (prepared by sol-gel method), or commercial $TiO_2$-P25 support via wet impregnation or photo-deposition. Ru particles prepared via the photo-deposition method were generally observed to be smaller than those prepared using the conventional wet

impregnation method. For instance, average sizes of $TiO_2$-P125 supported Ru particles prepared using the photo-deposition method were determined by TEM to be 0.8 nm, while those prepared using wet impregnation were found to have an average size of 2.9 nm. Similarly, the average sizes of Ca-modified-$TiO_2$ supported Ru particles prepared by the photo-deposition method were found to be 1.8 nm while those prepared using wet impregnation were 3.8 nm. The hydrogenation of LA to GVL was run using molecular $H_2$ and formic acid as a source of hydrogenation and $H_2O$ as a solvent at 190 °C.

Catalysts prepared via photo-deposition were observed to give higher GVL yields as compared to those prepared using wet impregnation. Of these catalysts prepared by photo-deposition, the Ca-based catalysts with small Ru particle (1.5 nm) catalysts were found to be the most active in the hydrogenation of LA to GVL. The 5% Ca-based catalysts displayed the highest LA conversion and GVL yield, irrespective of the catalyst preparation method used. However, when the most active catalyst (Ru/5% Ca-T500) is reduced at 200 °C, the average particle size was observed to increase and as a consequence, the GVL yield decreased from 50% to 20%. To investigate the effect of Ca loading on the performance of the catalysts, catalysts with different Ca loading (1% and 5% Ca) albeit with comparable Ru particle sizes (1.4 nm and 1.5 nm) were compared in the LA hydrogenation under similar reaction conditions. It was found that the 5% Ca catalyst displayed superior activity, rendering the concentration of Ca as another important factor influencing the activity of the catalysts. This was attributed to the fact that an increase of Ca resulted in a decrease of the anatase crystallite sizes, enhancing the dispersion of smaller Ru particles on the support surface. Moreover, the addition of $Ca^{2+}$ ions added to the titania support increased the basicity on the catalyst surface as well as metal–support interaction, which subsequently improves the hydrogenation activity of the Ca-based catalysts.

*2.4. Silica-Supported Ru Catalyzed Hydrogenation of LA*

One of the challenges in the hydrogenation of LA to GLV is that the product is in aqueous solution since GVL is miscible with $H_2O$. This necessitates the need for product separation after the reaction, which might be costly. In trying to overcome such costs and time-consuming processes, Poliakoff and co-workers reported the use of supercritical $CO_2$ for easy separation of pure GVL from $H_2O$ and unreacted LA [80]. In their work, 5% Ru supported on commercial $SiO_2$ (Degussa H 3036 XH/D) was used as a catalyst for the conversion of aqueous LA to GVL. This separation technique was inspired by the work reported by Lazzaroni et al. who added moderate supercritical $CO_2$ to THF/$H_2O$ which results in liquid/liquid separation into THF-rich and $H_2O$-rich phases [81]. Their reactor allowed them to drain unreacted LA while collecting water-free GVL product in the presence of $ScCO_2$ (see Figure 7). A GVL yield of 99% was obtained under optimum reaction conditions employed. Interestingly, they also claimed that their separation method managed to separate pure GVL even when the reaction is incomplete.

We have also reported on the use of silica-supported Ru nanoparticles as a catalyst in the hydrogenation of LA to GVL [75]. The silica-supported Ru particles, denoted as $Ru_{40}$@Meso-$SiO_2$, were determined to be well-dispersed and have an average size of $2.0 \pm 0.3$ nm by HRTEM. A LA conversion of 94% and 100% GVL selectivity was achieved for $Ru_{40}$@Meso-$SiO_2$ when using $H_2O$ as a solvent and optimum temperature and $H_2$ pressure of 150 °C and 10 bar, respectively. However, 100% LA conversion and GVL yield were observed when 1,4 dioxane was used as a solvent under similar conditions as those used for the $H_2O$ solvent. Some authors have reported Ru-$SiO_2$ (5% Ru loading) catalyzed hydrogenation of LA to GVL in ethanol and ethanol-water solvents [58]. In their case, higher LA conversion (98%) was observed for ethanol-water solvent as compared to that of the pure ethanol solvent (82.9%). However, both the selectivity and yield for GVL were observed to increase for reaction run in pure ethanol.

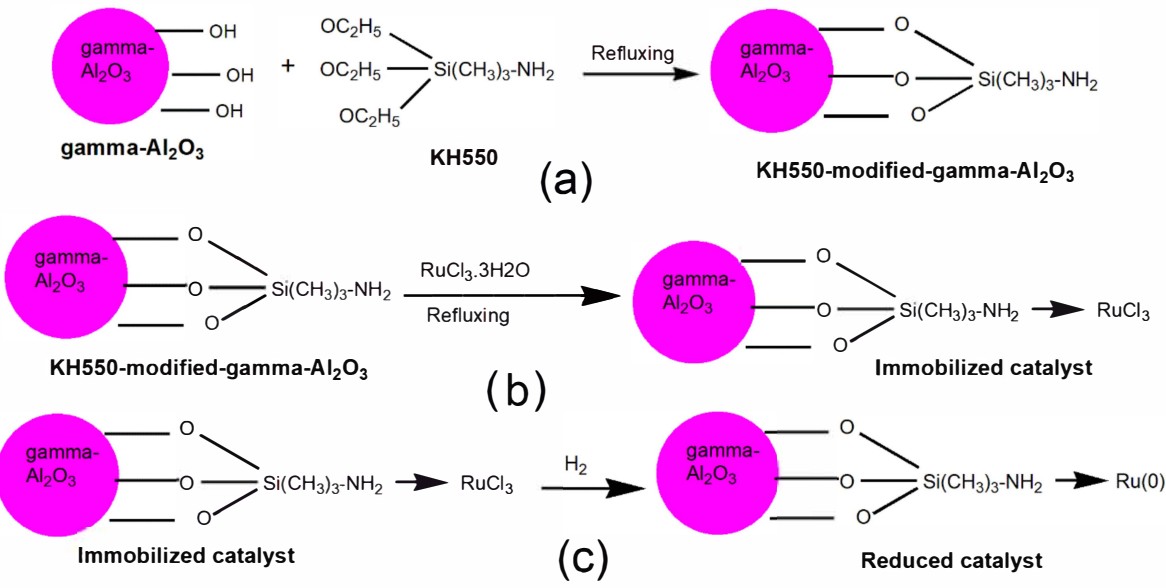

**Figure 7.** (**a**) Mechanism of surface modification by a silane coupling agent, (**b**) synthesis of the immobilized Ru-based catalyst, and (**c**) reduction of the immobilized Ru-based catalyst. Reproduced from ref [82] with permission from The Royal Society of Chemistry.

### 2.5. Zeolites and Other Oxides-Supported Ru Catalyzed Hydrogenation of LA

Another interesting comparison study in the catalyzed hydrogenation of LA using $Ru/ZrO_2$, $Ru/TiO_2$ and the commercial Ru/C catalysts was reported by Ftouni et al. [83]. The conventional wet impregnation method was applied for the synthesis of $Ru/ZrO_2$ and $Ru/TiO_2$ catalysts (1 wt% Ru). A wide range of analytical techniques such as TGA, BET, STEM, XPS, and XRD was used to characterize the as-synthesized catalysts before the evaluation in the hydrogenation of LA to GVL. The average diameter of Ru particles on fresh $Ru/TiO_2$ was determined by TEM to be $2.7 \pm 1.1$ nm. However, the average size of Ru particles on the $Ru/ZrO_2$ was not easily determined by TEM analysis.

All reactions were performed for 3 h at 150 °C, hydrogen pressure of 30 bar, and a stirring speed of 1250 rpm. Typically, a substrate (LA) to catalyst (bulk Ru) ratio of 1000 was used in the presence of a solvent (1,4-dioxane). Suitable standard reaction conditions for the investigation of the influence of the support on the catalyst performance were established using the commercial Ru/C catalyst. It was observed that the catalytic efficiency is highly sensitive to the reaction temperature and less sensitive to $H_2$ pressure. We, and other authors, have also observed a similar behavior of catalytic efficiency not showing many dependencies on the amount of $H_2$ pressure as opposed to temperature in the hydrogenation of LA to GVL [67,75]. Similarly, Yan et al. have reported that at a pressure above 12 bar of $H_2$, no increase in GVL yields was observed for reactions run in water [55].

When a reaction temperature of 100 °C was used, a full LA conversion was only observed after 24 h. However, at 50 °C, only less than 20% GVL was observed after a similar reaction time interval (24 h). A full LA conversion and quantitative GVL yields were observed after a reaction interval of 3 h and a temperature of 150 °C. As expected, the change in $H_2$ pressure from 20 to 40 bar at a fixed temperature (150 °C) was not found to show much effect on the overall outcome of the reaction as all different pressures investigated gave almost similar results. For instance, all pressure systems studied were reported to give nearly full GVL yields after 3 h reaction interval. Based on these preliminary findings, standard reaction conditions were set to at 150 °C, $P_{H_2}$ = 30 bar, and a fixed molar LA/bulk Ru ratio of 100. Using these reaction conditions, the commercial benchmark catalyst (Ru/C) was compared to that of the as-synthesized $Ru/TiO_2$ and $Ru/ZrO_2$ catalysts.

The amount of GVL produced gradually during the course of the reaction was investigated for all three Ru-based catalysts. A quantitative amount of GVL was observed for all three catalysts after 3 h. The TOFs values for Ru/C, Ru/TiO$_2$, and Ru/ZrO$_2$ were determined to be 0.53 s$^{-1}$, 0.29 s$^{-1}$, and 0.24 s$^{-1}$, respectively. Of the three catalysts, Ru/TiO$_2$ was observed to have changed its color from grey to dark black even after just one use. This, of course, signaling a possible deactivation of the Ru/TiO$_2$ catalyst during the reaction. This partial deactivation was confirmed by TEM and TGA post-characterization (for recycled catalyst) as neither as a result of the loss of metallic surface nor as a result of Ru sintering nor deposition of carbonaceous deposits on the catalyst's active sites. This partial deactivation was attributed to the modification of the TiO$_2$ (P125) support, instead. Furthermore, XPS analysis of the fresh and spent Ru/TiO$_2$ showed a significant difference in terms of Ti 2p species core level spectra. The noted change in color for the used Ru/TiO$_2$ catalyst was suspected to indicate the possibility of partial reduction of titania and subsequent formation of Ti$^{3+}$ species [84]. The significant amount of Ti$^{4+}$ that got reduced to Ti$^{3+}$ is said to have been caused by reduction conditions during LA hydrogenation. This reduction has been reported to be induced by Ru via "H$_2$ spillover" [85].

On the contrary, only a small change in terms of physicochemical properties was observed for the Ru/ZrO$_2$ even after five recycling tests. For instance, post-reaction catalyst characterization showed that the BET surface area did not change significantly, even after five reaction cycles. TGA analysis revealed that only 7% weight loss was observed after five reaction cycles. This weight loss was suspected to have caused carbonaceous deposits that resulted in a slight deactivation of the catalyst during recycling tests. Additional SEM analysis showed that both the fresh and the spent catalysts retained the same homogeneous size distribution within a few micrometers range (1–3 μm).

Multiple catalyst recycling tests were performed to assess the stability of the Ru/C, Ru/TiO$_2$, and Ru/ZrO$_2$ catalysts under applied batch conditions. Catalysts recovery between reaction cycles was achieved by filtering the solid catalyst, washing it with acetone, and leaving to dry overnight at 60 °C before the next reaction cycle. After five reaction cycles, the Ru/C catalyst displayed only a slight drop in GVL yield from 96 to 92% under the batch conditions applied. However, a clear deactivation for the Ru/C catalyst was observed for stability tests carried out at lower LA conversion (3 h at 100 °C). For instance, the GVL yield was found to drop from nearly 50% to 20% in the fifth recycling test. However, the selectivity towards GVL was observed not to change. The possibility of sintering causing this catalyst's deactivation was ruled off as TEM analysis for fresh Ru/C and the spent Ru/C (recycled five times) showed no difference in terms of particle sizes. For instance, the size of the average for fresh Ru/C and spent Ru/C (after five recycle) were determined to be 1.7 ± 0.4 and 1.7 ± 0.7 nm, respectively. BET measurements, however, revealed a decrease in surface area from 777 to 674 m$^2$/g after five reaction cycles. A similar trend was observed with pore volume as it dropped from 0.22 to 0.18 cm$^3$/g.

It only took three recycle runs for the Ru/TiO$_2$ catalyst to show some sign of deactivation with a drop of ~30% in GVL yield. Only a GVL yield of 63% was obtained after five reaction cycles. In the case of the Ru/ZrO$_2$ catalyst, a near qualitative yield of GVL was obtained up to the fourth reaction cycle, thereby proving to be highly stable. A slight drop in GVL yield, similar to that noticed for Ru/C catalyst at high LA conversion, was observed after the fifth recycling run. This slight deactivation was attributed to the carbonaceous deposits that may have built up during the catalyst recycling process. Recycling tests run for Ru/ZrO$_2$ catalyst at lower LA conversion also suggested higher stability for this particular catalyst.

Since dioxane has been deemed convenient when used on a small scale catalysis studies, the effect of other solvents was also investigated using various bio-based γ- and δ-lactones (tetrahydrofuran, γ/δ-hexalactone, γ-octalactone), including GVL itself. High GVL yields were observed with all γ-lactone solvents, including GVL itself. THF and δ-hexalactone proved less ideal for the reaction conditions applied and as a consequence, gave lower GVL yield, though with full selectivity towards GVL. This deactivation in

THF as a solvent was suspected to be resulting from THF polymerization into polyether polyol [86]. Post-run characterization of the catalyst by TGA analysis revealed about an 8% and 15% weight loss for δ-hexalactone and THF, respectively. Much less carbon deposition (≤5%) was observed for spent catalysts from runs in dioxane, GVL, γ-haxalactone, and γ-octalactone. TOF values for runs in GVL, dioxane, γ-octalactone were calculated to be 0.10, 0.24, and 0.26 s$^{-1}$, respectively. The difference in terms of catalyst performance between the δ- and γ-ketones was attributed to the corresponding enoic acid, which might be more prone to coking.

The influence of water on the performance of the catalyst was also evaluated using the Ru/ZrO$_2$ catalyst. This was simply carried out by adding a different amount of water to act as a co-solvent with the dominant dioxane. The addition of just 1% of water was found to bring an increase in GVL yield by 25% (after 1 h). When the amount of water was increased to 10%, a near-full LA conversion after 1 h resulted. However, when a large amount of water (more than 50%) was added, a slight drop in the activity was observed. This observation about the effect of water on the activity of the catalyst in the hydrogenation of LA to GVL correlates well with what other researchers have also reported [36,71].

Catalyst deactivation by potential impurities present in the feed reactor is a very important challenge to be considered for this catalytic hydrogenation of LA. The very same research group used the same set of catalyst (Ru/TiO$_2$ and Ru/ZrO$_2$) to study the effect of these impurities on the catalyst performance and stability for LA hydrogenation in both batch and continuous-flow systems [87]. The stability and performance of the catalysts were investigated using either water or dioxane as a solvent. For the study of the effect of impurities, experiments were performed in the presence and absence of impurities. For this purpose, impurities that include process-derived reagents (such as HCOOH, H$_2$SO$_4$, furfural (FFR), 5-hydroxymethylfurfural, humins) and biogenic impurities (such as sulfur-containing amino acids). For the performed impurities-free continuous-flow benchmark experiments, it was found that the Ru/ZrO$_2$ catalyst exhibit superior stability compared to the Ru/TiO$_2$ counterpart in dioxane solvent. On the contrary, the Ru/TiO$_2$ catalyst was observed to slightly show better stability than the Ru/ZrO$_2$ catalyst. The deactivation observed for Ru/TiO$_2$ in dioxane was attributed to the reduction of Ti$^{3+}$ and a decrease in surface area of titania as opposed to possible potential fouling or nanoparticles sintering.

The influence of impurities on the stability and performance of the catalysts was carried out in both batch and continuous-flow systems. The addition of impurities in the LA feed was found to affect each catalyst performance differently. For instance, the addition of HCOOH was found to cause a reversible deactivation for both catalysts. This was attributed to its preferential adsorption on Ru sites and possible CO poisoning. While on the other side, the presence of added impurities such as H$_2$SO$_4$, cysteine, and methionine all cause a permanent deactivation on both catalysts. When impurities such as HMF, FFR, and humins are used to mimic potential impurities in the LA feed, a gradual decrease in activity was observed for both solvents.

Some of the common problems of Ru/Al$_2$O$_3$ catalysts have been low activity and stability induced by the inhomogeneous dispersion of Ru particles and the unstable nature of Al$_2$O$_3$ in H$_2$O. Many other support materials including Al$_2$O$_3$ itself, TiO$_2$, MCM-41, SBA-15, and CeO$_2$ are unstable in an aqueous environment because of the existence of the hydroxyl groups (-OH) on their surfaces [88–94]. Tan et al. developed an integrated strategy for the design of highly active and stable Ru/Al$_2$O$_3$ catalysts [82]. The synthesis of this stable catalyst was achieved by modification of the abundant surface Al-OH groups of γ-Al$_2$O$_3$ with 3-aminopropyl triethoxysilane (KH550). This resulted in the transformation of the Al-OH groups into a stable Si-O-Si structure (see Figure 7). The amino ligands of KH550 were used to coordinate Ru active centers (with the electron-rich state) with the Al$_2$O$_3$ surface (r-Ru-NH$_2$-γ-Al$_2$O$_3$). TEM analysis showed that the prepared r-Ru-NH$_2$-γ-Al$_2$O$_3$ catalyst has Ru particles with a minimum average size of 1.2 nm, while that of unfunctionalized Ru/Al$_2$O$_3$ was determined to be 12.3 nm.

Both Ru/Al$_2$O$_3$ and r-Ru-NH$_2$-γ-Al$_2$O$_3$ catalysts gave the quantitative conversion of LA to GVL at 130 °C, after 30 min. For instance, complete LA conversion (100%) and higher selectivity for GVL was observed for both catalysts. However, at a lower temperature (70 °C), the r-Ru-NH$_2$-γ-Al$_2$O$_3$ catalyst showed superior performance as compared to the Ru/Al$_2$O$_3$ counterpart. The TOF value for the r-Ru-NH$_2$-γ-Al$_2$O$_3$ catalyst, for instance, was calculated to 3355 h$^{-1}$, about 8 times the TOF determined for Ru/Al$_2$O$_3$ catalyst (432 h$^{-1}$). It is noteworthy to mention that the r-Ru-NH$_2$-γ-Al$_2$O$_3$ catalyst was found to display excellent activity even at low temperatures such as 40 and 25 °C. For instance, LA conversion of 99% and GVL selectivity of 99.8% was achieved at 40 °C after 4 h. At room temperature, LA conversion of 99.1% and GVL selectivity of 99.9% was achieved, though after a longer reaction time (13 h). This excellent performance displayed by the r-Ru-NH$_2$-γ-Al$_2$O$_3$ catalyst in the hydrogenation of LA to GVL was attributed to the good dispersion of Ru particles on the KH500 modified Al$_2$O$_3$ surface. On the other hand, the activity of the Ru/Al$_2$O$_3$ catalyst was observed to decrease drastically at low temperatures. Only 39.5% LA conversion was recorded after 4 h, though the selectivity towards GVL did not change.

Although hydroxyapatite (HAP) supported catalysts (for metals such as Au and Ru) have been reported for other industrial reactions [95–97], the catalytic evaluation of HAP supported catalysts in the hydrogenation of LA to GVL was reported by Sudhakar et al. [98]. The HAP support was synthesized using co-precipitation, previously reported elsewhere by the same group [95]. Subsequent immobilization of metal catalysts (Ru, Ni, Pt, Pd) was carried out via wet-impregnation. The TEM characterization revealed Ru particles with a size range of 10–20 nm, for both unreduced and reduced Ru/HAP catalysts. As expected, the Ru/HAP (pre-reduced at 450 °C, 3 h) catalyst was observed to display superior activity in the hydrogenation of LA to GVL (70 °C, H$_2$ = 0.5 MPa, water solvent). LA conversion and GVL selectivity of 99% and 99% were obtained for Ru/HAP catalyst, respectively. However, the unreduced catalyst showed very poor activity with just 20% LA conversion, though with high selectivity towards GVL (99%). This observation, of course, suggests that metallic Ru are responsible for this hydrogenation activity for the reduced Ru/HAP. All other HAP-supported metals (Ni, Pt, Pd) gave LA conversion of less than 50% under identical reaction conditions.

The Ru/HAP catalyst was also used to investigate the influence of the solvent in the hydrogenation of LA to GVL. For this purpose, ethanol, ethanol + water (1:1 ratio), toluene, and water were all evaluated. It was observed that when ethanol (or ethanol: water) is used as a solvent at 70 °C, a significant amount of side products such as ethyl levulinate are formed. This is due to the transesterification of LA with ethanol. For instance, when ethanol and ethanol + water solvents were used, about 22 and 14% of ethyl levulinate was detected, respectively. However, it is noteworthy to highlight that no side products were observed when water was used as a solvent, and as such high LA (99%) and selectivity towards GVL (99%) were obtained. The conversion of LA was found to marginally decline by roughly 20% as the temperature is decreased to 60 °C, although maintaining high selectivity towards GVL. The reaction performed in the toluene solvent showed poor selectivity for GVL and this was attributed to the nonpolar nature of toluene. The recyclability test for Ru/HAP showed that the catalyst can be used for 5 reaction cycles without compromising selectivity for GVL. The conversion of LA was found to decrease by 10% from the first to the fifth reaction cycle.

A few years later, the same research group published an extended investigation of the same catalytic systems (HAP supported Ru, Pt, Pd, Ni, and Cu) in the hydrogenation of LA [99]. Ru/HAP was used to study the influence of temperature (275–425 °C) on the conversion of LA and product selectivity. The conversion of LA was found to increase as the temperature is elevated. For example, the conversion of LA increased from 65 to 94% as the temperature is raised from 275 to 425 °C (at GHVS = 3.89 mL·g$_{cat}$$^{-1}$·s$^{-1}$). However, the selectivity towards GVL was observed to decrease from 99.8 to 80% due to the formation of both α- and/or β-angelica lactones as side products. The influence of acidic sites on the catalysts' performance and selectivity during vapor phase LA dehydration was also

explored by the same research group using the same catalytic systems (Ru/HAP) [99]. It was suggested that at high temperatures, moderate acidic sites are contributing to the conversion of LA to angelica lactones. It was determined during the characterization of the Ru/HAP catalyst that it possesses a high-temperature $NH_3$ desorption peak attributed to strong acidic sites.

The effect of water in the catalytic performance of Ru/HAP was examined by carrying out the reaction using pure LA. As a consequence, only a small amount of LA was converted (4%) at 275 °C although maintaining 100% selectivity towards GVL. Of course, the positive influence of water on the hydrogenation activity has also been reported by others [36,71]. Interestingly, the selectivity for GVL and angelica lactone was observed to decrease when reactions were run using LA dissolved in other organic solvents such as methanol and ethanol. A significant amount of methyl levulinate formed as a result of the esterification of LA with methanol. It is also noteworthy to mention that role of acid sites on the hydrogenation of LA was investigated. For that purpose, Brønsted acidic sites of Ru were enhanced by the addition of dilute $H_2SO_4$ to the HAP support ($SO_4^{2-}$-HAP). Strong Brønsted acidic sites on the Ru/$SO_4^{2-}$-HAP were found to selectively catalyze dehydration of LA to angelica lactone. Based on these observations, it was, therefore, concluded that weak Brønsted acidic sites together with active metal sites are responsible for simultaneous hydrogenation and dehydration of LA to GVL.

Direct one-pot conversion of LA to GVL via hydrogenation using Ru nanoparticles embedded on sulfonic acid cation resin was reported by Moreno-Marrodan and Barbaro [100]. The as-synthesized Ru nanoparticles were immobilized on commercial sulfonic acid ion-exchange resin via wet impregnation yielding Ru particles with an average size of 2.8 ± 0.8 nm. The catalytic evaluation of the prepared resin supported Ru catalyst (Ru@DOWEX) was performed in both batch and continuous flow systems. The Ru@DOWEX was found to exhibit good activity for LA hydrogenation under mild conditions (T ≥ 50 °C, $H_2$ = 50 bar). For instance, full LA conversion and 99% selectivity towards GVL after 7 h.

The role of the presence of acidic sites of the support on the catalyst's activity was assessed by using corresponding non-acidic supports in which lithium ions substituted all mobile protons. A significant drop in the conversion of LA was observed when this non-acidic resin supported catalyst is used under identical reaction conditions used for acidic supports. The acidic catalyst was also evaluated in a continuous flow system. Preliminary results revealed that LA conversion of 90% with a GVL selectivity of ≥90%. Additionally, it was found that the conversion of LA can be improved to 100% either by decreasing the flow rate of the solution or by simply increasing the $H_2$ flow rate. The stability study of this catalyst showed that the catalyst can be reused over 3 consecutive runs without showing signs of deactivation. The excellent catalytic activity displayed by Ru@DOWEX was attributed to a combination of well-defined acid and RuNP active sites on the support.

The synthesis and catalytic evaluation (LA hydrogenation) of cross-linked sulfonated polyethersulfone supported Ru NPs (Ru/SPEP) has been reported [101]. The Ru/SPEP catalyst was prepared via wet impregnation using $RuCl_3$ and polyethersulfone as metal and support precursors, respectively. The formed supported Ru NPs were determined to have an average size of ~3.2 nm by TEM analysis. Preliminary catalytic investigation of the synthesized Ru/SPEP catalyst in the LA hydrogenation was carried out using mild conditions (70 °C, $H_2$ = 3MPa, 2 h). A conversion of LA of 87.9% and 99% selectivity towards GVL were obtained under those conditions. When Ru/$SiO_2$ and Ru/C were employed for comparison purposes, under similar reaction conditions, LA conversion of 22.2 and 54.5% were achieved, respectively. However, it is important to mention that both Ru/C and Ru/$SiO_2$ showed higher selectivity towards GVL (≥99.9%). It was also found that when the Ru/C or Ru/$SiO_2$ is blended with Amberlyst 15 (acidic), an improved LA conversion of 40.4% and 68.0% was achieved for Ru/$SiO_2$ and Ru/C catalysts, respectively, without compromising their selectivity towards GVL. These observations indeed confirm the role of acidic support in enhancing the performance of supported Ru catalyzed LA hydrogenation.

As a control experiment, the naked catalyst supports, SPES, also showed low conversion of LA, thereby confirming the role of acidic catalysts involving intermolecular lactonization and keto hydrogenation. To further confirm this, the acidic sites of Ru/SPES were neutralized by aqueous NaOH (Ru/SPESNa). The LA conversion was found to drop from 87.9 to 64.6% when the Ru/SPESNa catalyst is used. Blending this Ru/SPESNa catalyst with Amberlyst 15 yielded lower LA conversion (76.8%) as compared to that obtained for Ru/SPES (87.9%). Interestingly, it was found during the catalyst's stability study that the activity was increasing with the number of reaction cycles. For instance, LA conversion was observed to increase from 87.9 to 92.1% after the first hydrogenation cycle. This increase in activity as the number of reaction cycles increases was attributed to the surface RuO species formed when small Ru clusters are exposed to an oxygen-containing environment. These RuO species are said to possibly undergo gradual reduction during the course of the reaction, resulting in increased activity.

Another interesting study on the influence of aluminum coordination in zeolites-supported Ru catalyst was recently reported by Lu et al. [102]. This was motivated by other previous reports which suggested that the acidity of the zeolite has a strong dependency on the type of zeolite structure, the amount, and distribution of aluminum within the zeolite framework [103,104]. It was reported that the removal of aluminum from the zeolites (zeolite dealumination) framework can induce zeolite modification in terms of location, strength, number, and nature (Brønsted acid vs. Lewis acid site) of the acidic sites. This in turn can affect the catalytic properties of the zeolite supported catalysts in several ways [105].

The dealumination of the zeolite-supported catalysts (Ru/H-β and Ru/ZSM5) was investigated using a $^{27}$Al triple-quantum magnetic-angle spinning nuclear magnetic resonance spectroscopy (3QMAS NMR). It was observed during the LA hydrogenation that there was an increase in heterogeneity of aluminum speciation for both Ru/H-β and Ru/HZSM5. For the Ru/H-β catalyst, a severe loss of tetrahedral framework aluminum species (FAL) and extra-framework aluminum (EFAL) species into the liquid phase was observed. This consequently results in a significant decrease in strong acid sites as confirmed by FT-IR analysis. On the other hand, the symmetric FAL was mainly found to be converted into distorted tetrahedral FAL in the case of Ru/ZSM5 catalyst. Based on these observations, it was concluded that this decrease in acidity signal the inferior stability of the strongly acidic Ru/H-β as compared to Ru/ZSM5 counterpart under the applied condition. This was identified as the main reason why these two catalysts will display different catalytic performances as previously observed [72]. An interesting study has recently reported on the use of Ru supported on manganese octahedral molecular sieve (OMS-2) using a solvent-free method by Molleti et al. [106]. Under mild condition, the Ru/OMS catalyst managed to give LA conversion and GVL yield of 99.9 and 99.8%, respectively. The designed catalyst displayed both catalytic activity and stability even when reused four times.

The upgrade of LA to GVL using a Ru/C containing liquid triphase catalytic systems made of an aqueous phase, an organic phase, and an ionic liquid (IL) was reported by Selva et al. [107] (see Figure 8). Some of the most important benefits of these types of catalytic systems are: (i) easy separation of the product by a simple separation, (ii) preserving of catalyst's activity for in situ recycles without any metal loss, (iii) up to quantitative LA conversion with 100% GVL selectivity. In this catalytic system, the catalyst is stabilized inside different liquid phases from the reagents and reaction products and as a result, catalyst recovery becomes an easy process since the metal catalyst is often confined within the IL. The typical confinements of the commercial Ru/C within the IL in the triphase system was achieved by simply mixing Ru/C, [N$_{8,8,8,1}$][Cl] (IL1), H$_2$O, and isooctane (or ethyl acetate/acetone) until all the Ru catalyst was dissolved into the IL phase.

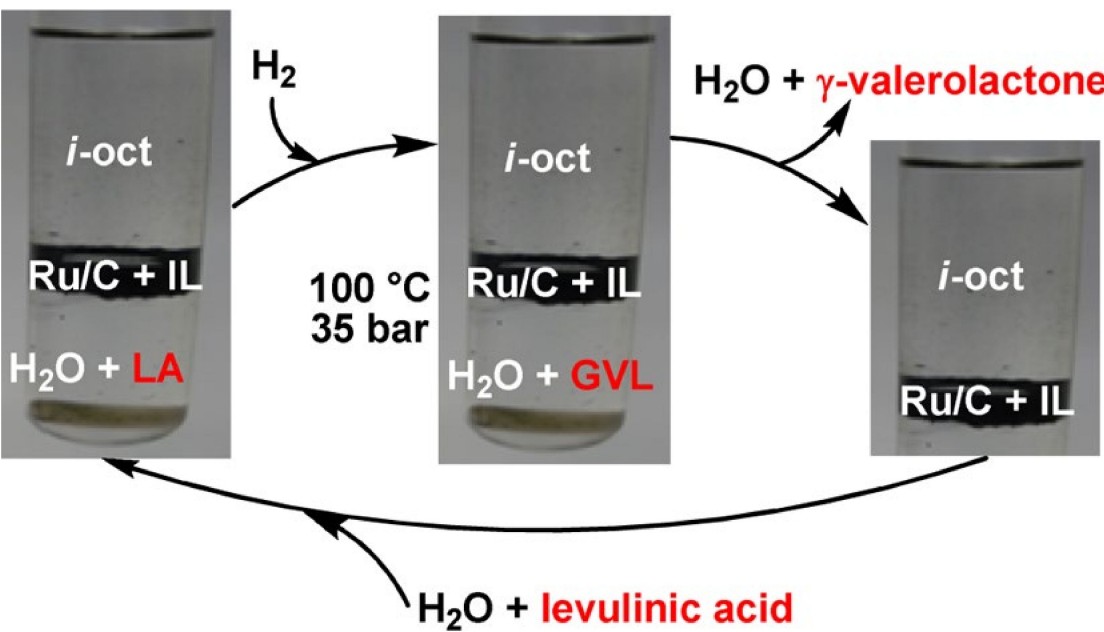

**Figure 8.** Pictures demonstrating the developed triphase catalytic system consisting of an aqueous phase, IL (confined Ru/C), and an organic phase. Reproduced from ref [107] with permission from American Chemical Society.

The preliminary hydrogenation reactions were initiated by adding LA into the tube containing the Ru/C-based triphasic catalytic system (ethyl acetate (EA), water and $[N_{8,8,8,1}][Cl]$). The tube was placed into the autoclave at the desired temperature (100 °C) and $H_2$ pressure (35 atm). In the absence of the IL, 100% LA conversion and 91% GVL selectivity were achieved. However, as IL is introduced into the catalytic systems, LA conversion was observed to drop drastically. This decrease in catalyst activity in the hydrogenation reaction was found to be more severe as the amount of IL is increased. For instance, when the IL/catalyst (wt/wt) ratio of 5.7 is used, the conversion of LA and GVL selectivity dropped to 56% and 55%, respectively. A further increase in the IL/catalyst ratio is increased resulted in a significant decrease in LA and GVL selectivity, giving 11% for both LA conversion and GVL selectivity. The separation of product, however, was almost impossible as the GVL was partitioned between EA and water in a 3:1 ratio. In trying to circumvent this separation challenge, a multiphase system consisting of an aqueous solution of LA, isooctane, 5% Ru/C, and $[N_{8,8,8,1}][Cl]$ was established. They referred to this setup as an "inverse multiphase" system because the organic reactant, LA, and the product, GVL remain dissolved in the aqueous phase. The role of isooctane in this setup was to bring good separation as well as catalyst segregation.

During the catalytic hydrogenation testing of this inverse multiphase catalytic systems, the optimum IL/catalyst (wt/wt) ratio was determined to be between the range of 3 and 6. Under the reaction conditions applied (100 °C, $H_2$ = 35 atm, 4 h), the conversion of LA in that optimum range was found to have improved to 60–65%. In all cases, however, GVL was observed as the sole main product. It was discovered, however, that the $[N_{8,8,8,1}][Cl]$ IL did form a distinct three-phase system later, as it was partially soluble in the aqueous phase. About 2–3% was determined to have been lost into the aqueous phase after each hydrogenation test run. In order to avoid that, less soluble IL liquids were prepared: $[N_{8,8,8,1}]$NTf IL2, $[P_{8,8,8,1}]$NTf IL3, $[N_{8,8,8,1}]$CF3COO IL4 and $[P_{8,8,8,1}]$NO3 IL5. Hydrogenation reactions were performed using triphasic catalytic systems made of different IL (IL1–IL5). A complete (100%) conversion of LA was achieved for all four salts IL (IL2–IL5), except for L1, which gave 32% LA conversion within 4 h. However, IL4 and IL5 were observed to be very soluble in the aqueous phase and as a consequence, were abandoned. On the other hand, IL2 and IL3 were stable enough not to partition into the

aqueous phase. Moreover, they both confined the Ru/C catalyst perfectly well without any detectable leaching.

The IL3 catalytic system was chosen for the investigation of the influence of the reaction time and ideal IL/catalyst ratio. For this purpose, three reaction sets were used (0.5, 1, and 2 h) were used. For each set of reactions, different IL/catalyst molar ratios, ranging between 6 and 25 to 1, were also used. Data obtained revealed that the amount of IL has very little (if any) effect on the LA conversion. However, an increase in LA conversion was observed as reaction time is increased. Most importantly, the multiphase catalytic system was found to show to remain even after 8 subsequent runs. This was attributed to the fact that the Ru/C remains well-confined even during catalytic reaction under the conditions applied.

### 2.6. General Discussion on Some Important Aspects: A Comparison Analysis

2.6.1. A Comparison with Other Metals (Other than Ru), the Effect of Solvent and Support in LA's Hydrogenation

Shown in Table 1 is a comparison of reported results obtained from different systems for the catalytic hydrogenation of LA to GVL using other metals than ruthenium as a catalyst. From all systems presented in Table 1, it is clear that the Ru-based catalysts have generally emerged as superior catalysts for the catalytic hydrogenation of LA irrespective of the solvent or support employed. In most cases a comparison in terms of the performance of Ru-based catalysts has been carried out using other noble metals such as Au, Pd, Pt, Ni. For instance, two independent comparison studies from the same research group have concluded that Ru catalysts are the best as compared to when other noble and non-noble metals such as Ni, Pt, Cu, or Pd are used for the same reaction under identical conditions. In one study (Table 1, entry 30–33), LA conversion and GVL selectivity of 99 and 99%, respectively, were reported for the Ru/HAP. On the other hand, the second highest LA conversion (for Pt/HAP) and GVL selectivity were reported to be 42 and 88%, respectively. A combination of more acidic/weak Bronsted and surface metal sites present on the Ru/HPA catalyst were suggested as the reason for this enhanced activity of the Ru catalyst. Similarly, in another related study by the same group, Ru displayed excellent activity for this hydrogenation of LA. Of all metals evaluated (Ru, Pt, Pd, Ni, Cu), the highest calculated TOFs ($2.90 \text{ s}^{-1}$) were for the Ru/HAP (Table 1, entry 25–29). The order of activity was determined was determined to be Ru > Pt > Pd > Ni > Cu.

A similar was also observed for another study involving titania-supported Ru, Au, and Pd (Table 1, entry 34–38). Again, Ru/TiO$_2$ catalyst was found to display excellent activity as compared to Au and Pd. Preliminary catalytic tests revealed that Pd- and Au-based catalysts were both almost inactive for the hydrogenation of LA, giving TOF values of $0.005 \text{ s}^{-1}$ and $0.004 \text{ s}^{-1}$. The Ru/TiO$_2$ catalyst, on the other hand, was observed to have completely converted LA even way before (40 min) the reaction set time (4 h), with a calculated TOF value of $0.5 \text{ s}^{-1}$. A higher dispersion of Ru over carbon support was suggested as the cause for superior catalytic performance of the Ru/C in another comparison study involving Ru, Pt, Pd metals (Table, entry 39–41). In this case, the Pt-based catalyst was found to have the lowest selectivity for GVL (30%). Other comparison studies involving ruthenium and other noble metals are presented in Table 1, entries 60–63, 42–44, and 16–18. In all those systems, Ru-based catalysts were observed to be the most active for the hydrogenation of LA to GVL.

Table 1. A comparison of Reported Results for Supported-Ru Catalyzed Hydrogenation of LA to GVL.

| Entry 1. | Catalyst | Solvent | Support | Reaction Conditions | LA Conversion (%) | Selectivity (%) | TOF ($h^{-1}$) | Ref |
|---|---|---|---|---|---|---|---|---|
| 1 | Ru | dioxane | Meso-$TiO_2$ | 150 °C, $H_2$ (10 bar), t = 5 h | 98 | 99 | 5023 | [75] |
| 2 | Ru | $H_2O$ | Meso-$TiO_2$ | 150 °C, $H_2$ (10 bar), t = 5 h | 92 | 98 | 3878 | [75] |
| 3 | Ru | dioxane | Meso-$SiO_2$ | 150 °C, $H_2$ (10 bar), t = 5 h | 100 | 100 | 6555 | [75] |
| 4 | Ru | $H_2O$ | Meso-$SiO_2$ | 150 °C, $H_2$ (10 bar), t = 5 h | 94 | 100 | 5314 | [75] |
| 5 | Pt | dioxane | Meso-$TiO_2$ | 150 °C, $H_2$ (10 bar), t = 5 h | 89 | 97 | 4780 | [75] |
| 6 | Pt | $H_2O$ | Meso-$TiO_2$ | 150 °C, $H_2$ (10 bar), t = 5 h | 88 | 98 | 4346 | [75] |
| 7 | Pt | dioxane | Meso-$SiO_2$ | 150 °C, $H_2$ (10 bar), t = 5 h | 84 | 97 | 5448 | [75] |
| 8 | Pt | $H_2O$ | Meso-$SiO_2$ | 150 °C, $H_2$ (10 bar), t = 5 h | 82 | 98 | 5112 | [75] |
| 9 | Ru | $H_2O$ | $TiO_2$ | 130 °C, $H_2$ (4 MPa), t = 3 h | 95 | 99.9 | | [71] |
| 11 | $Ru_{0.9}$-$Ni_{0.1}$ | Solvent-free | Ordered mesoporous carbons (OMC) | 150 °C, $H_2$ (4.5 Mpa), t = 2 h | 99 | 97 | 2501 | [64] |
| 12 | $Ru_{0.9}$-$Ni_{0.1}$ | Solvent-free | Carbon | 150 °C, $H_2$ (4.5 Mpa), t = 2 h | 34 | 15 | 426 | [64] |
| 13 | Ru | Solvent-free | Commercial carbon | 150 °C, $H_2$ (4.5 Mpa), t = 2 h | 48 | 30 | 870 | [64] |
| 14 | Pt | $H_2O$ | Hydroxyapatite(HAP) | 130 °C, $H_2$ (5 MPa), t = 12 h | 99 | 99 | – | [108] |
| 15 | Pt-Mo | $H_2O$ | HAP | 130 °C, $H_2$ (5 MPa), t = 12 h | 99 | 4 | – | [108] |
| 16 | Ru | $H_2O$ | $TiO_2$ | 70 °C, $H_2$ (50 bar) | 99 | 95 | – | [36] |
| 17 | Pt | $H_2O$ | $TiO_2$ | 70 °C, $H_2$ (50 bar) | 30 | 25 | – | [36] |
| 18 | Pd | $H_2O$ | $TiO_2$ | 70 °C, $H_2$ (50 bar) | 8 | 0 | – | [36] |
| 19 | Ru | $H_2O$ | $TiO_2$ | 70 °C, $H_2$ (4 MPa), t = 0.5 | 100 | 99.9 | 7676 | [71] |
| 20 | Ru | $H_2O$ | $ZrO_2$ | 70 °C, $H_2$ (4 MPa), t = 0.5 h | 92.1 | 99.9 | – | [71] |
| 21 | Ru | $H_2O$ | $SiO_2$ | 70 °C, $H_2$ (4 MPa), t = 0.5 h | 97.6 | 99.9 | – | [71] |
| 22 | Ru | $H_2O$ | $Al_2O_3$ | 70 °C, $H_2$ (4 MPa), t = 0.5 h | 73.3 | 99.9 | – | [71] |
| 23 | Ru | 2-sec-butyl phenol (SBP) | Carbon | 180 °C, $H_2$ (35 bar) | – | – | [a] 0.051 | [61] |
| 24 | RuSn | SBP | Carbon | 180 °C, $H_2$ (35 bar) | – | – | [a] 0.014 | [61] |
| 25 | Ru | $H_2O$ | HAP | 275 °C, $H_2$ (20 mL $min^{-1}$) | 92.2 | 99.8 | 2.90 | [99] |
| 26 | Pt | $H_2O$ | HAP | 275 °C, $H_2$ (20 mL $min^{-1}$) | 88.2 | 78.6 | 2.32 | [99] |
| 27 | Ni | $H_2O$ | HAP | 275 °C, $H_2$ (20 mL $min^{-1}$) | 21 | 65 | 0.19 | [99] |
| 28 | Pd | $H_2O$ | HAP | 275 °C, $H_2$ (20 mL $min^{-1}$) | 25.3 | 66.5 | 0.44 | [99] |
| 29 | Cu | $H_2O$ | HAP | 275 °C, $H_2$ (20 mL $min^{-1}$) | 32.2 | 74.8 | – | [99] |

**Table 1.** *Cont.*

| Entry 1. | Catalyst | Solvent | Support | Reaction Conditions | LA Conversion (%) | Selectivity (%) | TOF (h$^{-1}$) | Ref |
|---|---|---|---|---|---|---|---|---|
| 30 | Ru | $H_2O$ | HAP | 70 °C, $H_2$ (0.5 MPa) | 99 | 99 | – | [98] |
| 31 | Pd | $H_2O$ | HAP | 70 °C, $H_2$ (0.5 MPa) | 26 | 90 | – | [98] |
| 32 | Pt | $H_2O$ | HAP | 70 °C, $H_2$ (0.5 MPa) | 42 | 88 | – | [98] |
| 33 | Ni | $H_2O$ | HAP | 70 °C, $H_2$ (0.5 MPa) | 18 | 65 | – | [98] |
| 34 | Pd | dioxane | $TiO_2$ | 200 C, $H_2$ (40 bar), t = 40 bar | 2.5 | – | [a] 0.005 | [37] |
| 35 | Au | dioxane | $TiO_2$ | 200 °C, $H_2$ (40 bar), t = 4 | 3.6 | – | [a] 0.004 | [37] |
| 36 | Ru | dioxane | $TiO_2$ | 200 °C, $H_2$ (40 bar), t = 4 | 100 | 99 | [a] 0.5 | [37] |
| 37 | Au-Pd | dioxane | $TiO_2$ | 200 °C, $H_2$ (40 bar), t = 4 | 100 | 97.3 | [a] 0.1 | [37] |
| 38 | Ru-Pd | dioxane | $TiO_2$ | 200 °C, $H_2$ (40 bar), t = 4 | >99 | 99.6 | [a] 0.6 | [37] |
| 39 | Ru | dioxane | Carbon | 265 °C, $H_2$ (1 bar), t = 50 h | 100 | 98.6 | – | [66] |
| 40 | Pd | dioxane | Carbon | 265 °C, $H_2$ (1 bar), t = 50 h | 100 | 90 | – | [66] |
| 41 | Pt | dioxane | Carbon | 265 °C, $H_2$ (1 bar), t = 50 h | 100 | 30 | – | [66] |
| 42 | Ru | $H_2O$ | Carbon | 160 °C, $H_2$ (150 bar), t = 2 h | 100 | – | – | [65] |
| 43 | Pd | $H_2O$ | Carbon | 160 °C, $H_2$ (150 bar), t = 2 h | 70 | – | – | [65] |
| 44 | Pt | $H_2O$ | Carbon | 160 °C, $H_2$ (150 bar), t = 2 h | 53 | – | – | [65] |
| 45 | Pd-Re | $H_2O$ | Carbon | 160 °C, $H_2$ (150 bar), t = 2 h | 100 | – | – | [65] |
| 46 | Pt-Re | $H_2O$ | Carbon | 160 °C, $H_2$ (150 bar), t = 7 h | 100 | – | | [65] |
| 47 | Ru | dioxane | $Nb_2O_5$ | 200 °C, $H_2$ (40 bar), t = 4 h | 70 | 61 | – | [72] |
| 48 | Ru | 2-Ethylhexanoic acid (EHA) | $Nb_2O_5$ | 200 °C, $H_2$ (40 bar), t = 4 h | 90 | 95 | [a] 0.088 | [72] |
| 49 | Ru | dioxane | $TiO_2$ | 200 °C, $H_2$ (40 bar), t = 4 h | 100 | 96.8 | – | [72] |
| 50 | Ru | EHA | $TiO_2$ | 200 °C, $H_2$ (40 bar), t = 4 h | 100 | 99 | [a] 0.164 | [72] |
| 51 | Ru | dioxane | H-β | 200 °C, $H_2$ (40 bar), t = 4 h | 100 | 62 | – | [72] |
| 52 | Ru | EHA | H-β | 200 °C, $H_2$ (40 bar), t = 4 h | 100 | 85 | [a] 0.131 | [72] |
| 53 | Ru | dioxane | H-ZSM5 | 200 °C, $H_2$ (40 bar), t = 4 h | 100 | 51 | – | [72] |
| 54 | Ru | EHA | H-ZSM5 | 200 °C, $H_2$ (40 bar), t = 4 h | 100 | 85.5 | 0.204 | [72] |
| 55 | Cu | $H_2O$ | HAP | 275 °C, $H_2$ (mL min$^{-1}$) | 32.2 | 74.8 | – | [99] |

**Table 1.** *Cont.*

| Entry 1. | Catalyst | Solvent | Support | Reaction Conditions | LA Conversion (%) | Selectivity (%) | TOF (h$^{-1}$) | Ref |
|---|---|---|---|---|---|---|---|---|
| 56 | Cu | $H_2O$ | $ZrO_2$ | 200 °C, $H_2$ (500 psi), 5 h | 100 | 100 | – | [38] |
| 57 | Cu | MeOH | $ZrO_2$ | 200 °C, $H_2$ (500 psi), 5 h | 100 | 90 | – | [38] |
| 58 | Cu | $H_2O$ | $Al_2O_3$ | 200 °C, $H_2$ (500 psi), 5 h | 100 | 100 | – | [38] |
| 59 | Cu | MeOH | $Al_2O_3$ | 200 °C, $H_2$ (500 psi), 5 h | 100 | 86 | – | [38] |
| 60 | Pt | $H_2O$ | $TiO_2$(R10) | 70 °C, $H_2$ (50 bar), 1 h | 18 | 22 | – | [74] |
| 61 | Pt | $H_2O$ | $TiO_2$(R20) | 70 °C, $H_2$ (50 bar), 1 h | 23 | 27 | – | [74] |
| 62 | Ru | $H_2O$ | $TiO_2$(R10) | 70 °C, $H_2$ (50 bar), 1 h | 95 | 99 | – | [74] |
| 63 | Ru | $H_2O$ | $TiO_2$(R20) | 70 °C, $H_2$ (50 bar), 1 h | 100 | 100 | – | [74] |
| 64 | Ni | dioxane | $MgO_2$ | 160 °C, $H_2$ (3MPa), 1 h | 43 | <96 | – | [40] |
| 65 | Ni | dioxane | $MgO_2AlO_3$ | 160 °C, $H_2$ (3MPa), 1 h | 100 | 99.7 | – | [40] |
| 66 | Ni | dioxane | $Al_2O_3$ | 160 °C, $H_2$ (3MPa), 1 h | 26.1 | 96 | – | [40] |
| 67 | Ru | $H_2O$ | N-doped hierarchically porous carbon (NHPC) | 50 °C, $H_2$ (1 MPa), 3 h | >99 | >99 | – | [6] |
| 68 | Ru | $H_2O$ | Carbon | 50 °C, $H_2$ (1 MPa), 3 h | 50 | 60 | – | [6] |
| 69 | Ru | $H_2O$ | N-doped carbon sphere (N-CS-850) | 70 °C, $H_2$ (4 MPa), 1 h | 51 | 51 | 9858 | [109] |
| 70 | Ru | $H_2O$ | Carbon sphere (CS-850) | 70 °C, $H_2$ (4 MPa), 1 h | 32 | 32 | 6985 | [109] |
| 71 | Ru | $H_2O$ | Carbon | 90 °C, $H_2$ (45 bar), t = 6 h | 92 | 78 | – | [44] |
| 72 | Ru | $H_2O$ | $\gamma$-$Al_2O_3$ | 90 °C, $H_2$ (45 bar), t = 6 h | 26 | 62 | – | [44] |
| 73 | Ru | $H_2O$ | $TiO_2$ | 90 °C, $H_2$ (45 bar), t = 6 h | 26 | 54 | – | [44] |
| 74 | Ru | $H_2O$ | $\gamma$-$Al_2O_3$ | 70 °C, $H_2$ (4 MPa), t = 1 h | 25.7 | 99.8 | 432 | [82] |
| 75 | Ru | $H_2O$ | $NH_2$-$\gamma$-$Al_2O_3$ | 70 °C, $H_2$ (4 MPa), t = 1 h | 100 | 99.9 | 3355 | [82] |

[a] TOF expressed as per second (s$^{-1}$).

### 2.6.2. Effect of Support on the Performance of the Catalyst

Several studies have systematically investigated the effect of support on the performance of the catalysts for the hydrogenation of LA. A comparison of four different supports (Nb$_2$O$_5$, TiO$_2$, H-β, and H-ZSM5) in two different solvents, dioxane and EHA, in order to evaluate the effect of support (Table 1, entry 47–54). Higher selectivity towards GVL was observed for non-acidic supported catalyst (Ru/Nb$_2$O$_5$ and Ru/TiO$_2$) for reactions run in dioxane solvent. However, considerable amount of PA and its ester derivatives were observed for acidic zeolites supported catalysts Ru/ZSM5 and Ru/H-β). Of these acidic catalysts, the more acidic catalyst, Ru/ZSM5 gave higher yields for PA. Catalysts stability test run in dioxane suggested the Ru/TiO$_2$ catalyst is the most stable compared to other three catalysts. Higher selectivity for GVL were also observed for Ru/Nb$_2$O$_5$ and Ru/TiO$_2$ catalysts for reactions run in EHA solvent. The Ru/ZSM5 catalyst displayed superior activity than other catalysts (TOF = 0.24 s$^{-1}$). The acidic catalysts also showed higher selectivity for PA as compared to non-acidic catalysts for reactions carried out in AHA. However, PA yields obtained in EHA were lower than those observed in dioxane solvent.

Phase composition of the support has been reported to have an influence on the activity of the catalyst for the hydrogenation of LA. For instance, Pt-supported on titania composed of two different phase (rutile-anatase) phase (Table 1, entry 60–63). No clear correlation could be drawn for Pt supported on titatia with different phases. It is, however, noteworthy to mention that the Pt catalyst supported on pure anatase phase gave significantly higher GVL yields as compared to those supported on mixed-phases titania. On the contrary, the highest GVL yields (100%) were observed for Ru supported on mixed phase support. Strangely, lower GVL yields (38%) were observed for Ru catalysts supported on pure anatase phase. However, when the anatase is tuned by calcination, the activity was observed to tremendously improve, with LA conversion of 99% and GVL yields of 99% and 93%, respectively. Ru catalysts were generally observed to display a superior activity as compared to Pt counterparts under similar conditions. This superiority shown by Ru-based catalysts was attributed to what was already explained by Michael et al. [36]. It is clear that the nature of the titania can affect the activity of the metal differently. The different in the catalytic performance of Ru-based catalysts on different titania phases was explained by two key properties: (i) electronic properties, which govern the dinging of metal and the support and its dispersion of the oxide support, (ii) morphology, which ultimately impact on the surface area and morphology. It was determined during TEM analysis that Ru particles seem to prefer to grow on the rutile surface. The same, however, was not observed for Pt catalysts, which were observed to be randomly distributed on the two phases (anatase and rutile). The difference in electronic properties of the rutile and the anatase phase of the TiO$_2$ has been identified as the reason for weaker adhesion of Ru particles on the anatase phase by a difficult electron transfer from the metal to semiconductor. While Pt, on the other hand, was determined not to show any strong preference in terms of adhesion to either of the TiO$_2$ phases.

The functionalization of the carbon support with nitrogen has been found to enhance the catalytic performance of the Ru supported catalysts. Ru supported on N-dope porous carbon (Ru/NHPC) was observed to exhibit more stability for the aqueous hydrogenation of LA (Table 1, entry 67–68). This enhanced activity of the Ru/NHPC catalyst was attributed to its high solubility in H$_2$O as compared to the Ru/C catalyst. This catalyst was found to be very stable even after reused for 13 times. While on the other hand, a significant decline in activity for the Ru/C was observed. Similarly, an enhanced catalytic performance was reported for Ru supported on N-doped carbon spheres (Table 1, entry 69–70). The presence of the quaternary N atom was cited as the main reason for good dispersion of Ru nanoparticles on the carbon sphere (Ru/N-CS-850). The Ru/N-CS-850 observed to display excellent stability even after 4 reuse cycles. A clear sintering was seen for the unfunctionalized Ru/C catalyst. The presence of strong acid sites and elevated temperatures (325–425), other side products such as angelica lactones (AL) can also be observed at the expense of GVL yields. For instance, the increase in temperature from 275 to

425 C was found to increase the conversion of LA from 64 to 94%. However, the selectivity towards GVL was observed to drop from 99.8 to 80.5%. This was due to the formation of other side products (AL).

Titania-supported Ru catalyst has been reported to display more stability than those supported on different supports ($SiO_2$, $Al_2O_3$, and $ZrO_2$) (Table 1, entry 19–22). This stability was attributed to a better dispersion of Ru nanoparticles on $TiO_2$, which subsequently facilitates the generation of Ru active sites are selectively active for this hydrogenation reaction. On the contrary however, $ZrO_2$ has been reported to be ideal catalyst support as compared to titania ones [83,87]. However, this stability was found to show dependency on the reaction media. In both cases from two independent studies reported by the same research group, have concluded that $ZrO_2$ supported catalyst proved more robust than the $TiO_2$ and carbon supported counterparts for reactions run in dioxane solvent. Better activities for $TiO_2$ supported Ru catalysts in dioxane have also been reported somewhere else (Table 1, entry 1–8). In another case, the $TiO_2$ was observed to slightly show more stability than the $ZrO_2$ supported catalyst. The deactivation of the $TiO_2$ support during recycling test was attributed to the reduction of $Ti^{4+}$ species to $TiO^{3+}$, leading to a decrease in surface area.

### 2.6.3. Effect of Bimetallic Catalysts on the Overall Catalyst Performance

The use of bimetallic catalysts in the catalytic hydrogenation of LA has also been reported to bring some stability and enhanced catalytic activity. For example, the incorporation of Re to the already active Ru/C was found enhance the catalyst performance and stability [59]. In the presence of the sulfuric acid (a catalyst in the conversion of cellulose to obtain LA), the activity of the Ru/C was found to decrease. The selectivity for GVL decreased from >98% to 60%. However, in the case of the bimetallic RuRe/C, the activity and stability were maintained even in the presence of sulfuric acid. In fact, slightly higher activities were observed for RuRe/C in the presence of sulfuric acid as compared to when there is no acid added. In another related study, the addition of Sn to the Ru/C was found to increase the catalyst's stability (Table 1, entry 23–24). In this case, as the amount of Sn added to the Ru/C is increased, the activity of the resulting bimetallic was found to decrease the hydrogenation rates. Just a small addition of Sn to the Ru/C catalyst resulted in the decrease in the activity of the catalyst. Although the rates observed for monometallic Ru/C catalyst were 4–5 times higher than any bimetallic tested, it was found to suffer from deactivation. During this deactivation of the catalyst over time, the LA hydrogenation selectivity was found to increase. Additionally, the deactivation process was irreversible. For the bimetallic RuSn catalyst at the atomic ratio of Sn:Ru to 4:1 and beyond, the resulting bimetallic catalyst was found to be inactive for this reaction under the condition employed. Bimetallic with higher Sn loading did not show any sign of deactivation even after long reaction hours in a continuous system. This observation indicated that the bimetallic RuSn catalysts are stable even in the presence of other impurities.

Another stable bimetallic of Ru/Ni metals supported on ordered mesoporous carbons (OMC) has been reported (Table 1, entry 11–13). A high TOF value of 2501 $h^{-1}$ was observed for the Ru/Ni-OMC catalyst as compared to the TOF value (2288 $h^{-1}$) determined for the monometallic counterpart, Ru/OMC. This of course, indicates the enhancement exhibited by bimetallic systems. This TOF value for monometallic Ru/OMC catalyst, was however, much higher than the one reported for Ru supported on commercial carbon (870 $h^{-1}$). When the same bimetallic catalyst with the same ratio is prepared using carbon as a support, a drastic drop in both LA conversion (34%) and GVL yields (15%) was observed, indicating the role played by the mesoporous channels in enhancing the activity and selectivity. The Ru/Ni/OMC bimetallic catalyst was observed to retain its stability after 15 reuses. This robust nature displayed by bimetallic catalysts was attributed to the enhanced, strong interaction of the immobilized bimetallic Ru-Ni NPs with the mesoporous carbon support. A similar enhancement induced by the presence of bimetallic system using Re as a promoter metal for Pd- and Pt-based catalysts (Table 1, entry 42–46). For instance, the LA conversions

for monometallic Pd/C and Pt/C catalysts were observed to improve from 70 and 53% to 100 and 100%, respectively, when bimetallic Re-promoted catalysts were used (Pd-Re/C and Pt-Re/C).

2.6.4. The Effect of Solvent on the Hydrogenation of LA to GVL

Just like other factors, the choice of a solvent has been observed to influence the performance of the catalyst performance/stability as well as the selectivity of the product/s. The performance of Ru/C was evaluated in different alcohols as solvents, namely: methanol ethanol and 1-butanol [58]. The highest yield and selectivity for GVL was observed when methanol. These results were comparable to those obtained for dioxane solvent. The worst performance was observed when the reaction was run in ethanol solvent. It should be mentioned though that all solvents showed higher selectivity for GVL (81–85%). Higher solubility of $H_2$ in methanol as opposed to other two alcohols was cited for this enhanced activity in methanol. The addition of to the ethanol and butanol solvents resulted in the improved activity. For instance, the addition of 10% (by volume) to the ethanol and butanol solvent significantly increased the conversion of LA and selectivity, signaling the enhanced activity induced by the presence of $H_2O$.

The use of 2-sec-butylphenol (SBP) has been preferred in other cases in order to recover LA from aqueous solutions of sulfuric acids which are usually present in the LA feed. However, this SPB solvent is also prone to hydrogenation. The hydrogenation of SPB solvent was suppressed by the addition of significant amount (equimolar) of Sn to the Ru/C catalyst (for the fabrication of RuSn/C catalyst). In some instances, a solvent has been used as a source of hydrogen donor. A study by Xu et al. [78] reported the use of 2-propanol as a solvent and source of internal hydrogen for CTH production of GVL from methyl levulinate (ML) under continuous flow system. The Ru/TiO$_2$ catalyst gave an excellent, achieving ML conversion and selectivity of 98 and 97%, respectively. Another study on the CTH conversion of LA, EL, or ML substrates to GVL has been reported using different alcohols (methanol, ethanol, 1-propanol, cyclohexanol, 2-butanol, and 2-propanol) as internal H donors over Ru(OH)$_x$/TiO$_2$ catalyst [76]. The highest conversion and GVL yield were observed when 2-propanol solvent was used. Under similar reaction conditions, moderate GVL yields could be obtained when 2-butanol and cyclohexanol were employed as both solvents and H donors. The excellent catalytic performance displayed in the 2-propanol solvent was attributed to high adsorption of 2-propanol on the Ru(OH)$_x$/TiO$_2$ catalyst. Al other primary alcohols (methanol, 1-propanol, ethanol) did not show any ability to act as internal hydrogen donor. This attributed to the difficulty in β-hydride elimination from primary alcohols [77]. Excellent GVL yields were also observed when 2-propanol is used as an internal H-donor for conversion of either butyl levulinate (BL) or EL substrate. Interestingly, GVL yields were observed to increase as the metal loading of the catalyst is increased (from 0.8 to 4 mol%). GVL was scarcely produced when low metal loading (0.8 mol% Ru) is used. This was attributed to strong adsorption of LA on the Ru(OH)$_x$ sites (for higher metal loading) and possibly due to generated water molecules generated during the reaction.

A comparison between $H_2O$ and THF solvents in the catalytic conversion of LA to GVL using three different metals (Ru, Pd and Pt) has also been reported [36]. Both Ru and Pd catalysts did not show any activity in THF solvent. Small, yet significant LA conversion and GVL yields were observed for Pt/TiO$_2$ in both solvents. Ruthenium was observed to display superior activity in $H_2O$, achieving LA conversion and GVL yield of 99 and 95%, respectively. The increase in activity of the Ru catalyst was attributed to the hydrogen bond effect.

The product selectivity has been observed to be influenced by the choice of solvent too (Table 1, entry 47–54). Non-acidic supported catalysts (Ru/Nb$_2$O$_5$ and Ru/TiO$_2$) gave highest LA conversion and GVL yields in dioxane solvent. In the case of acidic zeolite supported Ru catalyst, side products such as PA and its ester derivatives were observed in considerable amount. Agglomeration of Ru particles was observed for Ru/TiO$_2$,

and zeolites-supported Ru catalysts. Additionally, post-characterization of spent catalysts revealed that the $Ru/Nb_2O_5$ and zeolites supported catalysts suffer from decreased in surface area and pore volume due to carbon residue/coke on the surface of the catalysts. These observations are, of course, an indication of how selectivity can differ for catalysts possessing different acidity. In the 2-ethylhaxanoic acid (EHA), all catalysts showed good catalytic performance. The acidic Ru/ZSM5 catalyst was, however, found to be the most active (based on calculated TOF values). Again, some selectivity towards the formation of PA was observed for both acidic catalysts. However, the yields of PA in EHA were observed to be lower than those obtained in dioxane. These observations, of course signify the competition that exist between LA and GVL (prior to PA formation) with the solvent for adsorption with the catalyst.

Another comparison study on the effect of water and other organic solvents (ethanol and toluene) also selected $H_2O$ as an ideal solvent for the aqueous hydrogenation of LA to GLV [98]. Fairly good conversion and selectivity were observed for ethanol and ethanol + $H_2O$ solvents. However, significant amount of ethyl levulinate was observed as a result of transesterification of LA with ethanol. These side products were not observed when $H_2O$ was used as solvent, even at different temperatures. Very poor GVL yield was seen for toluene solvent. This was attributed to the non-polarity nature of this solvent. This promotional catalytic effect of water as a solvent has also been reported by Tan et al. [71]. The addition of water to (67 wt%) to the reaction system was found to drastically accelerate the conversion of LA (43.8 to 100%) while maintaining the selectivity towards GVL (99.9%). Other employed organic solvents (methanol, ethanol, and dioxane) either showed low LA conversions or lower selectivity for GVL. However, when water is added to these solvents, both LA conversion and GVL selectivity improved significantly. This observation confirms the positive role of water in the aqueous hydrogenation of LA to GVL. The ability of water to transport H atoms on the catalyst surface and the possibility of participation of water itself in the reaction. The enhanced activity observed in the presence of water solvent has been reported even for non-noble catalysts (Table 1, entry 56–59).

GVL has itself, been used as a solvent in the hydrogenation of LA to GVL over Ru–Sn/C catalyst [63]. In this study, the GVL is used to extract LA and FA from aqueous solutions. The biphasic phase between H2O and GVL can be created by tuning the aqueous phase. This is carried out by the addition of certain solutes such as salts and sugars to the aqueous phase, allowing LA to partition into the organic phase. The advantages of using GVL as a solvent for this reaction are: (i) no need for separating product with the solvent as the product is the solvent itself, (ii) this approach eliminates the deposits from solid humin species formed during decomposition step of cellulose. In another interesting study, GVL was screened with other organic (dioxane and THF) as well as bio-based γ-(valerolactone, hexalactone) and δ-hexalactone lactones for use solvent for the catalytic hydrogenation of LA to GVL over $Ru/ZrO_2$ catalyst [83]. All γ-lactones, including GVL were found to give high GVL yields. THF and δ-hexalactone gave the lowest GVL yields under similar conditions, although maintaining high selectivity for GVL. This reduced activity observed in THF solvent was attributed to the a possible THF polymerization into polyether polyol [86]. The effect of water was also investigated by adding different amounts of water to the dioxane solvent. For comparison purposes, the anhydrous dioxane. As expected, water was found to have positive impact in the hydrogenation activity. For instance, just an addition of 1% of water to the dioxane solvent was observed to increase GVL yield by 25%.

### 2.6.5. Effect of Promoters on Ru Catalysts

There are many factors that have been reported to promote the performance of the Ru-supported catalysts for the hydrogenation of LA to GVL. For example, the alloying of Ru-based with a suitable metal (such as Re, Pd, Sn) to form the bimetallic catalytic system has been discovered to enhance or promote the performance/activity of the monometallic Ru-based catalyst (see Section 2.6.3). The promotion effect on Ru-based catalysts has also

been observed with the addition of a co-catalyst. Galletti et al. showed a promoted effect on the performance of carbon or alumina-supported Ru catalysts by introduction of co-catalyst (such as ion exchange resin Amberlyst A70 or a15, niobium oxide or phosphate) in the reaction systems (see Section 2.2).

The addition of base to the $Ru(OH)_x$-based catalytic hydrogenation of ML to GVL can also have some promoting effect on the catalyst performance as demonstrated by Kuwahara et al. [76]. For instance, the addition of MgO or HT to the reaction system was found achieve almost full conversion of ML and higher selectivity for GVL, with a GVL yield of 98% obtained. In that way, the formation of 4-hydroxy pentanoic acid methyl ester (4-HPM) was suppressed in favor of GVL formation. This observation was attributed to prominent basicity of the added solid bases which will subsequently usually promote the dealcoholation step of 4-HPM to form GVL. When other bases such as $Na_2CO_3$ and trimethylamine (TEA) are added as additives, a drop in the ML conversion was observed, without altering GVL yield. Recently, a study on the enhanced activity of the $Ru/TiO_2$ catalyst by addiction of calcium (alkali metal) was reported [79]. The investigations on the effect of addition of Ca on the catalytic performance was carried out using various techniques. The Ca-modified supported catalyst was observed to display higher activity. This was due to the formation of small, well-dispersed Ru nanoparticles on the newly formed (at the expense of anatase crystallite sizes) titanate phase. This results in the strong interaction between small Ru nanoparticles and the modified titania support. Additionally, the addition of Ca for surface modification is associated with the increase in concentration and strength of basic sites. The presence of these basic sites was cited as another reason for the enhanced catalytic performance for Ca-modified $Ru/TiO_2$ catalysts.

### 2.6.6. Challenges on Catalyst' Stability

Although, many heterogeneous-based catalysts have been reported for the hydrogenation of LA to GVl. The development of a stable, highly active heterogeneous catalysts for the production of GVL from biomass-derived molecules remain an ongoing challenge. Often, a catalyst deactivation is observed during the course of the reaction, rendering it not idea to use for continuous flow system used for large-scale industrial production of GVL from LA. This catalyst deactivation result from a number of factors such as metal leaching, coke formation (mostly for carbon support), change in textural properties of the support, or oxidation/reduction of support material. In the study reported by Yang and Lin, the Ru/C was found to display excellent activity for the hydrogenation of LA to GVL [55]. However, a drastic decrease in both conversion and GVL selectivity was observed just after one recycle run in methanol solvent. A possible metal leaching was suggested for this decrease in activity. In other studies, a stable Ru/C catalyst was prepared by addition of acid co-catalyst, ion exchange resin Amberlyst A70. No loss in activity was observed for this catalyst even after 5 recycles for longer reaction time (30 h). In some cases, the nature of solvent also plays a role in the stability of the catalyst. For instance, some Ru leaching was observed for $Ru/TiO_2$, Ru/H-β, Ru/ZSM5 catalyst in dioxane solvent, with Ru/ZSM5 showing the highest metal loss in dioxane. Under similar conditions, these catalysts displayed much improved stability for reactions run with EHA solvent [72].

In another study, using the benchmark Ru/C catalyst, a continuous deactivation of the catalyst was observed during stability test experiments [69]. A combination of metal leaching (about 7.4% after first run) and a slight decrease on the surface were correlated to the instability shown by Ru/C catalyst. On the contrary, analogous Ru catalyst supported on $ZrO_2@C$ ($Ru/ZrO_2@C$) was shown to exhibit excellent stability even after 6 consecutive runs. In the presence of added HCl in the reactor feed, to mimic conditions employed for industrial production of GVL from LA, the Ru/C was completely inactive after the first run, owing to its poor stability under harsh conditions. A drastic decrease in surface was observed for Ru/C under acidic media. This observation was attributed to a possible build of carbonaceous deposits, formed during acid catalyzed process. On the other hand, they showed no sign of deactivation even under this acid

media. Excellent stability displayed by the Ru/ZrO$_2$@C catalyst was attributed to strong interaction between Ru and ZrO$_2$@C support, which subsequently result in the formation of well dispersed active Ru species. The observation of course, signifies the role of choice of support on the development of highly stable catalysts. The same research group has also reported a study that showed the ability sulfates (in the presence of sulfuric acid impurities) adsorbed on ZrO$_2$ support to protect active Ru species from deactivation [110]. The partial deactivation of support materials in some cases has resulted in the deactivation of the catalyst. For instance, the Ru/TiO$_2$ was found to show quick deactivation during recycling experiments as compared to the Ru/ZrO$_2$ counterpart [83]. This deactivation was attributed to the reduction of the titania support in combination with a partial coverage of active Ru nanoparticles due to a detrimental strong metal–support interaction. The t Ru/ZrO$_2$ catalyst was observed to possess a robust character as it did not show any deactivation sign even after 4 consecutive recycle runs. In other cases, the impurities (discussed in Section 2.5) in the LA feed can play a role in the deactivation of the catalyst. A recent study reported a study comparing the stability between Ru/TiO$_2$ and Ru/ZrO$_2$ catalyst in the presence of added impurities. The Ru/TiO$_2$ catalyst was found to deactivate during extended times on stream. Post catalyst characterization revealed that this deactivation for titania supported catalyst was due to reduction of the titania surface as well as the decrease in surface area. The same could not be said when it comes to the Ru/ZrO$_2$ catalyst. It is noteworthy to mention that, in the absence of added impurities, the Ru/TiO$_2$ catalyst showed slightly higher stability than Ru/ZrO$_2$ in water solvent. Again, these observations indicate the influence the choice of support can have for this catalytic conversion of LA to GVL. In other cases, the reduction or hydration of support surface groups which may result in catalyst deactivation was suppressed or circumvented by functionalization of the support with heteroatoms such as N has proved promising in designing stable, active catalysts for the transformation of LA to GVL [6,109].

### 2.6.7. Vapor Phase Hydrogenation of LA to GVL

A number of studies have investigated the catalytic conversion of LA to GVL over Ru-based (and other metals) catalysts under vapor phase conditions [29,61,66,99]. The use of vapor phase for the conversion of LA to GVL offers a number of advantages over the liquid phase process: (i) no need for high pressure conditions, (ii) no need or little product purification is required, (iii) minimizes catalyst deactivation and offers easy catalyst recycling process, and (iv) greatly minimizes the generation of waste associated with the use of organic solvents. For instance, at a fixed temperature of 265 °C, Ru/C catalyst was observed to give a full LA conversion irrespective of the amount of pressure used (1–25 bars) while still maintain high selectivity for GVL [66]. This activity for Ru/C was maintained for 10 days with only about 25 decrease in selectivity towards GVL. Good dispersion of Ru over carbon support was suggested as the reason for the excellent performance displayed by Ru/C under vapor phase condition. We, and others, have also observed a minimal effect of H$_2$ concentration for reactions run at 150 °C, though in liquid phase process [67,75]. The performance of the commercial Ru/C was also shown to have a strong dependency on the reaction temperature but did not show any sensitivity to the amount of H$_2$ pressure [83]. A full LA conversion was obtained at 100 C after 24 h. At 50 °C, only 20 GVL yield was achieved within the same time interval. A full LA conversion and quantitative yields of GVL was achieved within 3 h.

### 2.6.8. Advancement on the Development of New Method

There has been an increase from scientists around the globe in devoting most of their research focus on the development of catalytic processes for the conversion of biomass-derived molecules (such as LA) to values-added chemicals (such as GVL) and other important liquid hydrocarbons. Of these, most published work has put more focus on the development of a stable catalyst. However, there are few exceptional articles that deserve to be highlighted as they immensely contribute in process designing of this catalytic conver-

sion of these bio-based molecules into biofuels, fuel additives, and other valuable chemicals. For example, the one-pot conversion of cellulose to hydrocarbon fuel [1]. In their approach, a cascade method was employed. This approach progress through the removal of oxygen from biomass while at the same facilitate the separation of products. The decomposed solid cellulose (in aqueous sulfuric acid) produces equal mixture of LA and FA. The FA subsequently decomposes into $H_2$ and $CO_2$. This formed $H_2$ can then be used for the conversion of LA to GVL over Ru/C. The GVL is then passed over the sulfur-tolerant nobia-supported Pd catalyst to give PA, which subsequently get converted via ketonization to give 5-nanonone (See Figure 3).

In another related study, LA was converted into a range of valuable chemicals and fuels using few reaction and product separation steps [24]. As in their other study, LA is converted in the presence of Ru/C catalyst to give GVL. GVL was hydrogenated by passing it over a nobia-supported Pd catalyst to produce PA. The selectivity to PA was found to be sensitive to metal loading of the catalyst. Lower metal loading (>1%) was found to promote the formation of PA. This increased yield of PA helps in achieving reduced content of oxygen from LA. At elevated temperatures, this very Pd/Nb$_2$O$_5$ catalyst can still be used for the conversion of PA to 5-nonanone. This opened a great possibility for direct conversion of GVL to 5-nonanone.

A novel study on the use supercritical in order to achieve a complete separation of product during Ru/SiO$_2$ catalytic hydrogenation of LA [80]. This separation was achieved by pumping LA + $H_2O$ through a heated reactor containing $CO_2$ and $H_2$. The resulting pure GVL product can be collected from the separator. No detectable amount of LA could be measured from the collected GVL. However, a detectable amount of unreacted LA was observed in the aqueous stream. This energy efficient process eliminates the purifying process by distillation associated with high costs.

The stability test at lower LA conversion has been found to give a clear indication about the stability of the catalyst. For example, has been shown to display excellent stability for batch systems even after five reaction cycles [83]. However, when the same catalyst is test for stability at lower LA conversion, a clear deactivation was observed in the fifth run. The GVL yield was observed to have dropped from nearly 50% to 20% for lower LA conversions experiments performed.

An important utilization of ionic liquid in the enhanced separation of the product during LA conversion has been highlighted [107]. In this approach, a triphase system comprised of aqueous phase, organic phase, and an ionic liquid was developed for use in the selective catalytic conversion of LA to GVL over Ru/C. Using this innovative design set-up, a quantitative amount of LA was converted with 100% selectivity towards GVL. Most importantly, the product was recovered by a simple separation. Additionally, the recovered catalyst, which remained confined during the reaction by ionic liquid, did not show signs of metal leaching and hence preserved activity.

## 3. Perspective and Conclusions

GVL is an important molecule that can be obtained from the catalytic hydrogenation of biomass-derived molecule, LA. From the afore-discussed Ru catalyzed hydrogenation of LA to GVL systems, it is without any doubt that, indeed heterogeneous Ru catalysts are emerging as superior catalysts for this important reaction. In some instances, the catalysts with more Lewis-acidic sites were found to favor the formation of PA and ring-opening of GVL. Alloying Ru catalysts with other metals such as Sn and Ni seem to significantly improve catalyst's activity, stability and product selectivity. Interestingly, catalytic hydrogen transfer (CHT) using other sources of hydrogen than molecular $H_2$ has been established. For example, alcohols such as isopropanol/isobutanol have successfully been used as alternative hydrogen donor source for this hydrogenation of LA to GVL [76]. Similarly, formic acid has also been used as an internal source of hydrogen. The use of co-catalysts (such as A15 and A70) with the Ru catalyst has been observed to result in enhanced hydrogenation activity.

Of great importance was the potential displayed by water as a potential "green" solvent for the aqueous hydrogenation of LA to GVL. Water was found to enhance the hydrogenation activity when used as an alternative solvent. Modification of support (e.g., $Al_2O_3$ with KHOO, C with N, and $TiO_2$ with Ca) improves metal–support interaction, catalyst stability, and subsequently the activity. In other cases, it was established that a synergetic effect exists between support and Ru particles. Additionally, textural properties of the support material also play a role in the catalyst's activity. $ScCO_2$ can be used to solve problems associated with the separation of GVL from $H_2O$ and other unreacted molecules in the end of the reaction cycle. Confinement of the supported catalyst by ionic liquids proved to greatly preserve the catalyst' activity. Interestingly, $ZrOr_2$ support, though unusually used as catalyst' support for this reaction, has been found to be more robust than catalyst than the benchmark $Ru/TiO_2$ catalyst.

Based on the discussion above, it is only fair to say there is a satisfying progress in the search for developing stable, active, and selective catalysts for the conversion of biomass derived compounds to bio-based fuels and other value-added chemicals. However, it is also worthy to mention that few systemic studies are still missing in the literature. For instance, a number of studies focused on carbon and titania supported Ru-based catalysts as benchmark catalysts. Only recently, $ZrO_2$ is emerging as an alternative superior catalyst support for Ru nanoparticles. There is still room for development of other cheaper, greener (renewable), stable supports for the conversion of bio-based molecules into biofuels and chemicals. Additionally, it would be interesting to have more studies based on the use of novel metal organic frameworks for this reaction. Most of heterogenous catalytic systems have been reported to show strong some dependency on the size of the nanoparticles, however, sufficient systematic studies are still lacking for these systems. A satisfying number of reports based on innovative reactor design that utilize critical $CO_2$, ionic liquid, and vapor phase processes for efficient separation of GVL product is still lacking. More literature on the solvent-free continuous systems would accelerate the development of greener processes for this particular reaction.

**Author Contributions:** M.M. drafted and edited the manuscript. S.M. provided the funds and edited the manuscript. R.M. provided supervision and edited the manuscript. All authors have read and agreed to the published version of the manuscript.

**Funding:** This work is based on the research supported by the National Research Foundation of South Africa through the DST/NRF Research Chair in Biofuels (UID 91635).

**Conflicts of Interest:** Authors declare no conflict of interest.

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
