# Peer review of "Heterogeneous Ru Catalysts as the Emerging Potential Superior Catalysts in the Selective Hydrogenation of Bio-Derived Levulinic Acid to γ-Valerolactone: Effect of Particle Size, Solvent, and Support on Activity, Stability, and Selectivity"

_catalysts, doi:10.3390/catal11020292_

Round 1

Reviewer 1 Report

The authors present the review of the recent advances in the selective hydrogenation of bio-derived levulinic acid to γ-5 valerolactone by heterogeneous Ru catalysts.

The review is well structured and written.

However the quick literature check returns two recent articles, one recent review article and one recent book chapter in the field of this review:

https://doi.org/10.3389/fchem.2020.00221

http://dx.doi.org/10.3390/catal10060692

https://dx.doi.org/10.1021/acssuschemeng.9b07678

http://dx.doi.org/10.1021/bk-2020-1359.ch008

Please update the literature and the discussion of the manuscript.

Author Response

The authors present the review of the recent advances in the selective hydrogenation of bio-derived levulinic acid to γ-5 valerolactone by heterogeneous Ru catalysts.

The review is well structured and written.

However the quick literature check returns two recent articles, one recent review article and one recent book chapter in the field of this review:

https://doi.org/10.3389/fchem.2020.00221

http://dx.doi.org/10.3390/catal10060692

https://dx.doi.org/10.1021/acssuschemeng.9b07678

http://dx.doi.org/10.1021/bk-2020-1359.ch008

Response: We are grateful for this suggested literature papers by the reviewer. The discussion on the manuscript has been updated based on all literature suggested by the reviewer. All suggested references have been included in the “introduction” section and “section 1.4” of the manuscript

Reviewer 2 Report

Review attached

Author Response

Review of "Heterogeneous Ru Catalysts as the Emerging Potential Superior Catalysts in the

Selective Hydrogenation of Bio-Derived Levulinic Acid to γ-Valerolactone: Effect of Particles

Size, Solvent, and Support on Activity, Stability, and Selectivity" by Mulisa Maumela et al.

The manuscript of Mulisa Maumela et al. aims to review recent advances in the conversion of levulinic

acid (LA) to γ-valerolactone (GVL), with a focus on Ru-based catalysts and the effect of catalysts

structure and reaction conditions on catalytic activity, stability and selectivity. While the topic is

relevant to the catalysis community, the review does not have a clear structure, the figures do not convey

the main message of the review and the content is not critically discussed.

I therefore think the manuscript might be suited for publication in catalysts after major restructuring

and rewriting. My comments follow.

  1. The authors should provide a list of other similar/relevant reviews on the topic of biomass

upgrading, to help the reader finding relevant information (e.g.

https://www.aimspress.com/fileOther/PDF/energy/energy-07-02-165.pdf,

https://www.sciencedirect.com/science/article/pii/S1385894719309842) Moreover, the

authors should address this very similar review about Ru catalysts, and discuss what the added

value of their work is: https://www.frontiersin.org/articles/10.3389/fchem.2020.00221/full

Response to reviewer: We appreciate this important suggestion from the reviewer. The discussion including suggested paper has been added under “introduction section” of the manuscript.

  1. The authors should include a critical discussion about the fact that Ru is a platinum group metal,

and is included in the 2020 list of critical materials (https://eurlex.

europa.eu/legalcontent/EN/TXT/HTML/?uri=CELEX:52020DC0474&from=IT). Can we

find alternatives to Ru (“synthetic” Ru), e.g. by using alloys of more available metals? If not,

why, what is missing on the catalyst design/mechanism elucidation part?

Response to reviewer: We appreciate this input from the reviewer. It is unfortunate that we were unable to open the link provided here. However, some information on the ruthenium being listed in the group of platinum metals has been included towards the end of the introduction part.

  1. The review is now organized mainly by supports (see chapter 1.2 on). However, the content of

the chapters does not correspond with the title. In each section, the authors present many relevant

parameters, such as solvent, conditions, bimetallics… This results in a scattered information,

with many repetitions and some contradicting remarks (e.g. line 181 vs 416-418, where water

is said to have two opposite H2 solubilities). I strongly recommend to avoid this division in terms

of “supports”, and restructure the paper according to the next comments.

Response to reviewer: We totally agree with the suggestion by the reviewer. The manuscript has been updated according to what the reviewer suggested 

  1. The introduction reports a lot of details about lignocellulosic conversion (e.g. lines 55-77), while

the focus on Ru, LA and GLV is lost. It would be good to have here a discussion about the

relevance of LA and GLV (e.g. market volumes, costs, LCAs, CO2 reduction impact…), and

about Ru when compared to other catalysts. Also here the scope of the literature review should

be clarified, the similar reviews shortly listed and finally a brief overview of the sections of the

review given.

Response to reviewer: More discussion on Ru, LA and GVL has been added in the introduction section as per reviewer’s suggestion.

  1. The figures are not explicative of the content or focus of the review. Especially Figures 4 (bad

resolution), 5, 6, 7. Figures representing comparison of Ru with other metals, bimetallic effects,

solvent effects and so on should be included to help the reader visualize the main take-home

messages.

Response to reviewer: Figure 4 and 7 have been removed from the manuscript. More description has been added for Figure 5 and 6. Table 1 has been added to compare Ru with other metals.

  1. Following are some sections that I identified along the review (line number in brackets) that

address the same topic and should be rearranged and discussed together:

  • Other metals: A comparison among Ru, Pt, Pd, Au and other metals should be made,

in a systematic and critical way, with a table or a summarizing figure, based among

others on ref. 30, 33-37, 49. (287-299) Re, Pd, Pt, (784-797) Ru, Ni, Pt, Pd, (811-817)

Ru, Pt, Pd, Ni, Cu. How does the metal influence activity and chemoselectivity?

Response to reviewer: We appreciate this important suggestion. Section 1.5.1. has been included to address these suggestions.

  • Support effects: similarly, a concise but systematic overview of the effect of supports

should be given in a figure or a table. Which is the optimal support (ZrO2 emerges as

best candidate but only in the conclusions now) and why is it the best? What is the role

of acidity, the preservation of surface area under reaction conditions, the better

dispersion of the active phase? (334-368) supports, reduction effect, inclusion of N in

the support, and then other metals are discussed all together here, (446-459) TiO2,

Nb2O5, ZSM-5, (501-540) TiO2 phases, (page 16, alternating with solvent effect, phase

and loading effects), (page 18, with deactivation details that may be discussed

separately together), (754-782) structure of support and its surface modification,

Response to reviewer: We appreciate this important suggestion. Section 1.5.2. has been included to address these suggestions

(818end of page 23) acid sites of supports, acidity effect.

  • Bimetallics: the reason for enhanced performances of bimetallics should be critically

discussed. Is it electronic effects, geometric effects such as ensemble control, or higher

dispersions? (209-225) RuRe – but also the role of H2SO4 can be separately discussed,

(229-234) RuSn, (259-277) RuNi, (430-445) RuPd, ensemble control.

Response to reviewer: We appreciate this important suggestion. Section 1.5.3. has been added to address these suggestions

  • Solvent effects: there are many solvents properties to be taken into account when

designing LA upgrading processes. These should be reviewed systematically, but are

now spread out all over the manuscript: e.g. (170-184) H2 solubility, (234-258)

extraction of LA and FA, and humin solubilization in GVL, (280-283) water effect,

(390-end of page), (460-500) solvent effect on surface area and support stability,

adsorption of La, GVL and solvent, solvent-free reactions, (572-588) solvent hydrogen

donor capability, and beta H elimination, (page 18, together with T, P conditions effect),

(728-753) role of solvent on stability, (798-810) transesterification of solvents, taking

part in the reaction.

Response to reviewer: We appreciate this important suggestion. Section 1.5.4. has been included to address the effect of solvent

  • Promoters: the main promoters of Ru catalysts should be discussed. Now only Ca is

included (page 17). What is the effect of alkali metals addition, and how does this

correlate with the importance of support acidity?

Response to reviewer: We appreciate this important suggestion. A discussion on the promoters of Ru has been added under Section 1.5.5

  • Vapor phase: what are the benefit/downsides of not using solvents? (300-333) the effect

of T and atmosphere composition should be discussed more clearly, (676-687) T, P

effects on Ru/C.

Response to reviewer: We appreciate this important suggestion. Section 1.5.6. has been included to address these suggestions

  • Stability/deactivation: the main reasons for deactivations should be addressed: loss of

surface area, leaching, carbon deposition, etc. (134-162) Ru leaching, (371-376)

deactivation.

Response to reviewer: We appreciate this important suggestion from the reviewer. Section 1.5.7. has been included to address the stability/deactivation issues

  • Process/method developments: some remarks on advancements in reactor engineering

and separation should be grouped together: (190-199) cascade method, (200-208)

5nonanone, (625-637) supercritical CO2, (711-724) catalysts recycling and recovery,

could be discussed more critically, (900-end) ionic liquids.

Response to reviewer: Section 1,5.8.

  1. The authors should finally include a perspective section, in which the critical points to be

addressed in Ru-based catalysts for biomass upgrading are discussed and future research

directions can be suggested.

Response to reviewer: A perspective section has been incorporated with the “Conclusion” section under section 1.6.

Round 2

Reviewer 2 Report

I thank the authors for considering my comments and improving the manuscript accordingly.

I can now recommend accepting the manuscript in the present form.